# PhyloGen: Language Model-Enhanced Phylogenetic Inference via Graph Structure Generation

**Chenrui Duan**[1,2*] **Zelin Zang**[2*] **Siyuan Li**[1,2] **Yongjie Xu**[1,2] **Stan Z. Li**[2†]

[1]Zhejiang University, College of Computer Science and Technology;   [2]Westlake University

duanchenrui@westlake.edu.cn;

{zangzelin; lisiyuan; xuyongjie; stan.zq.li}@westlake.edu.cn

[*]Equal contribution   [†]Corresponding author

## Abstract

Phylogenetic trees elucidate evolutionary relationships among species, but phylogenetic inference remains challenging due to the complexity of combining continuous (branch lengths) and discrete parameters (tree topology). Traditional Markov Chain Monte Carlo methods face slow convergence and computational burdens. Existing Variational Inference methods, which require pre-generated topologies and typically treat tree structures and branch lengths independently, may overlook critical sequence features, limiting their accuracy and flexibility. We propose PhyloGen, a novel method leveraging a pre-trained genomic language model to generate and optimize phylogenetic trees without dependence on evolutionary models or aligned sequence constraints. PhyloGen views phylogenetic inference as a conditionally constrained **tree structure generation** problem, jointly optimizing tree topology and branch lengths through three core modules: (i) Feature Extraction, (ii) PhyloTree Construction, and (iii) PhyloTree Structure Modeling. Meanwhile, we introduce a Scoring Function to guide the model towards a more stable gradient descent. We demonstrate the effectiveness and robustness of PhyloGen on eight real-world benchmark datasets. Visualization results confirm PhyloGen provides deeper insights into phylogenetic relationships.

## 1 Introduction

Phylogenetic trees [42] (or evolutionary trees) are tree-structured graphs representing kinship relationships between species [11], where each leaf node represents a distinct species and internal nodes represent evolutionary bifurcations. The tree's topology reflects evolutionary relationships based on their genetic characteristics, and branch lengths indicate evolutionary distances. Phylogenetics study is the foundation of evolutionary synthetic biology and is critical for tracking the evolutionary trajectories of species, analyzing the transmission pathways of newly discovered viruses, and providing meaningful insights for clinical applications [25, 13, 33].

**Background.** Despite DNA sequencing technologies [32] allowing us to construct evolutionary relationships based on molecular properties (DNA, RNA, and proteins), phylogenetic inference remains challenging due to its complex parameter space, which encompasses both continuous (branch lengths) and discrete (tree topology) components. Traditional MCMC-based methods like MrBayes [29] and RevBayes [9], despite their ability to extensively explore the huge tree space, are hindered by slow convergence rates. Additionally, the number of possible tree topologies for n species grows factorially as $(2n-5)!!$ for $n \geq 3$, posing huge computational challenges. In contrast, VI-based methods leverage approximate inference to provide more efficient estimations. Depending on the data type and research objectives, these methods can be further categorized into tree representation learning and tree structure generation. **Tree Representation Learning**, as shown in Fig. 1(a),

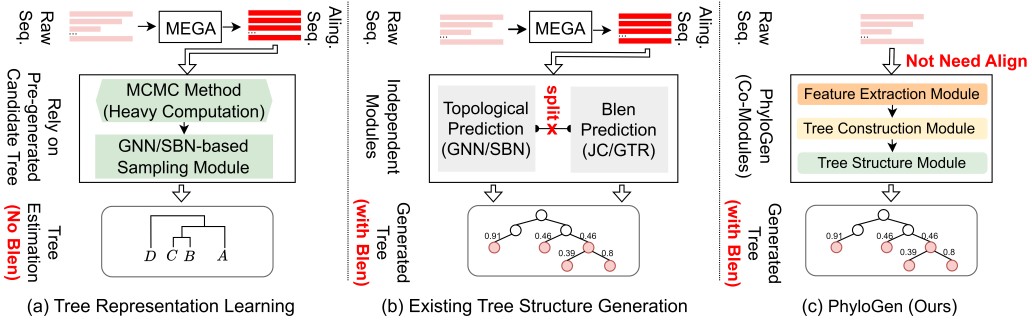

Figure 1: **Comparison of PhyloTree Tree Inference Methods.** **(a)** The inputs are aligned sequences, and topologies are learned from existing tree structures using methods like SBNs, which rely on MCMC-based methods for pre-generated candidate trees without considering branch lengths directly. **(b)** The inputs are aligned sequences, and then tree structures and branch lengths are directly inferred by variational inference and biological modules. These methods optimize tree topology and branch lengths separately. **(c)** The inputs are raw sequences processed by a pre-trained language model to generate species representations. Then, an initial topology is generated through a tree construction module, and the topology and branch lengths are co-optimized by the tree structure modeling module.

intends to learn topological representations from existing tree structures. For instance, Subsplit Bayesian Networks (SBNs) [45] focus on probabilistic representations from given tree structures without considering branch lengths. Variational Bayesian Phylogenetic Inference (VBPI) [46] and its extensions VBPI-GNN [44] utilize the topological probabilities provided by SBNs and jointly model tree structures in latent space by deriving branch lengths through variational approximations. However, these methods require pre-generated topology, based on which better topological representations are then learned. **Tree Structure Generation**, as shown in Fig. 1(b), aim to infer tree structures directly from sequences. PhyloGFN [48] integrates VI and reinforcement learning to construct tree topologies, simplifying branch lengths into discrete intervals, thus necessitating posterior data for inference. VaiPhy [16] introduces the SLANTIS sampling strategy and basic biological models (e.g., Jukes-Cantor(JC) model [24]) for topology and branch length estimation. ARTree [36] builds tree topologies recursively using a graph autoregressive model. GeoPhy [22] models tree topologies in a continuous geometric space. However, these methods require input sequences to be aligned to equal lengths, and tree topology and branch lengths are optimized separately. They also ignore sequence features that are critical for accurate phylogenetic analysis.

**Our Method.** As depicted in Fig. 1(c), we propose a novel approach based on a pre-trained genome language model. Our model does not rely on evolutionary models or the requirement to align input sequences to equal lengths and fully exploits the prior knowledge embedded in biological sequences. PhyloGen models phylogenetic tree inference as a conditional-constrained tree structure generation problem, aiming to generate and optimize the tree topology and branch lengths jointly. We map species sequences into a continuous geometric space and perform end-to-end variational inference without restricting topological candidates. To ensure the topology-invariance of phylogenetic trees, we incorporate distance constraints in the latent space to maintain translational rotation invariance. Our approach demonstrates effectiveness and efficiency on the eight real-world benchmark datasets and verifies its robustness through data augmentation and noise addition [41]. In addition, we propose a new scoring function to guide the model towards a more stable and faster gradient descent.

## 2 Related Works

**MCMC-based methods**, such as MrBayes [29] and RevBayes [9], have been widely used for phylogenetic inference due to their ability to explore vast tree spaces.

**VI-based methods** offer a more efficient alternative to MCMC by leveraging approximate inference techniques. These methods can be categorized into two main approaches: structure representation and structure generation. **Tree Representation Learning.** This approach focuses on extracting information from existing tree structures. SBNs [45] capture the relationships between existing

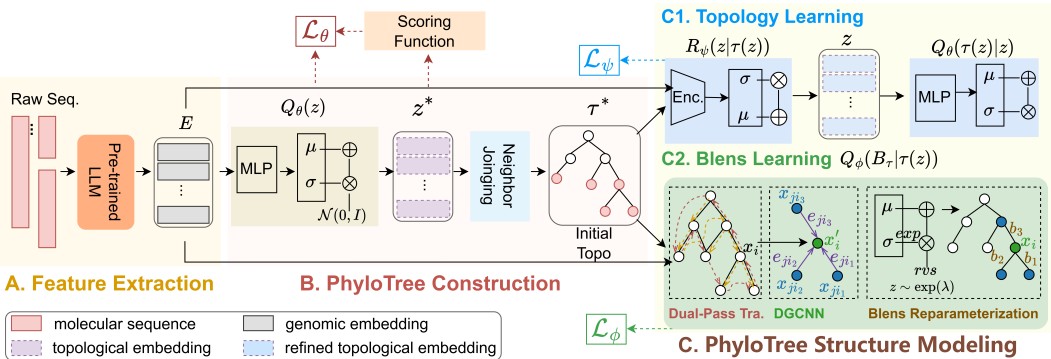

Figure 2: **Framework of PhyloGen. A. Feature Extraction** module extracts genome embeddings $E$ from raw sequences $Y$ using a pre-trained language model. **B. PhyloTree Construction** module uses $E$ to compute topological parameters, which generate an initial tree structure $\tau^*$ via the Neighbor-Joining algorithm. **C. PhyloTree Structure Modeling** module jointly model $\tau$ and $B_\tau$ through the topology learning component (TreeEncoder $R$ and TreeDecoder $Q$) and the branch length (Blens) learning component (dual-pass traversal, DGCNN network, Blens reparameterization).

subsplits without addressing branch lengths. VBPI [46] and its variants VBPI-NF [43] and VBPI-GNN [44]: These methods introduce a two-pass approach to learn node representations, including branch lengths. VBPI employs variational approximations to handle branch lengths, allowing for joint modeling of the tree's latent space. **Tree Structure Generation.** This approach aims to infer tree structures directly from sequence data. PhyloGFN [48] utilizes GFlowNet [10, 20, 21], a combination of VI and reinforcement learning, which requiring posterior data for accurate inference. VaiPhy [16] incorporates a sampling strategy and basic biological models to estimate topology and branch lengths. ARTree [36] employs a graph autoregressive model to build detailed tree topologies. GeoPhy [22] models tree topologies within a continuous geometric space, offering a different approach to the distribution of tree topologies. For details on related work, please see Appendix B.

## 3 Methods

**Notation.** The phylogenetic tree is denoted as $(\tau, B_\tau)$, where $\tau$ is an unrooted binary tree topology reflecting evolutionary relationships among $N$ species. $B_\tau$ denotes the non-negative evolutionary distances of each branch. The tree consists of $N$ leaf nodes, each corresponding to a species, and several internal nodes. PhyloGen aims to generate tree topology and branch lengths directly from the raw sequences.

**Framework.** We model the phylogenetic tree inference problem as a tree structure generation task under conditional constraints, consisting of three main modules as shown in Fig. 2. Feature Extraction module extracts genome embeddings from raw sequences via a pre-trained language model. PhyloTree Construction module uses these embeddings to generate an initial tree structure using a tree construction algorithm, introducing the latent variable $z^*$ to represent the tree topology. PhyloTree Structural Modeling module iteratively refines the tree structure and branch lengths through topology learning and branch length learning components. By integrating these modules, we transform the complex dynamics of evolutionary history into a tree-based learning framework, facilitating a deeper understanding of phylogenetic relationships.

**A. Feature Extraction**   We utilize DNABERT2 [49], a genome language model, to transform molecular (DNA) sequences $Y = \{y_i\}_{i=1}^N$ into genomic embeddings $E = \{e_i\}_{i=1}^N$. These embeddings discern complex patterns and long-range dependencies, serving as the basis for generating the latent variables $z^*$, which dynamically inform both the topology and the branch lengths. By introducing DNABERT2, we redefine the construction of phylogenetic trees as a continuous optimization problem within a biologically meaningful embedding space.

**B. PhyloTree Construction**    To construct the phylogenetic tree, genomic embeddings $E$ are input into an MLP network to derive the parameters $\mu$ and $\sigma$ of the latent space, representing topological embeddings $z^*$. This latent space effectively captures the evolutionary relationships that guide the subsequent tree structure generation process. The latent variable is then generated using the reparameterization trick [15]: $z^* = \mu + \sigma \odot \varepsilon$, $\varepsilon \sim \mathcal{N}(0, I)$. Then, distance matrix $D$ is computed using $z^*$: $D(i,j) = \sum_{i,j=1}^{N} z_i^* \oplus z_j^*$, where $\oplus$ means an XOR operation reflecting the nucleotide mismatches. This distance matrix is fed into the Neighbor-Joining (NJ) algorithm [31] to generate the initial tree topology $\tau(z^*)$. The tree's topology is structured using a parent-child relationships list $[L[i], R[i], P[i]]$, where $P[i]$ denotes the parent of node $i$, and $L[i]$ and $R[i]$ denote the left and right child node. This representation highlights the tree's hierarchical nature, optimizing its structure based on the evolutionary patterns in the data.

**C. PhyloTree Structure Modeling**    The purpose of the tree structure modeling module is to jointly optimize both the phylogenetic tree's discrete topology and continuous branch lengths.

**C.1. Topology Learning.**    The first component involves a TreeEncoder $R(z|\tau(z^*))$ and a TreeDecoder $Q(\tau(z)|z)$ to learn the tree topology. Concretely, the encoder $R$ receives an initial tree topology $\tau(z^*)$ and genomic embeddings $E$ from DNABERT2 as inputs, conditions the latent state $z$ to refine topological embeddings in a continuous space. We introduce variational inference to enhance the quality of the tree structure and adapt to data uncertainty and complexity by minimizing the Kullback-Leibler (KL) divergence. Additionally, the encoder $R$ acts as a regularization mechanism, reducing overfitting and enhancing gradient stability. To refine the topology, the decoder $Q$ samples from the probability distribution parameterized by the encoder's latent state. This process optimizes model parameters by minimizing the KL divergence between the true distribution $P(\tau(z)) = P(y|\tau(z), B_\tau)$ and the variational distribution $Q(\tau(z)) = q(\tau(z), B_\tau)$. To facilitate backpropagation through stochastic nodes, the latent variable $z$ is sampled using the reparameterization trick: $z = \mu + \sigma \odot \varepsilon$, where $\varepsilon \sim \mathcal{N}(0, I)$ introduces controlled stochasticity while maintaining differentiability, ensuring that $z$ effectively captures refined topological embeddings.

**C.2. Branch Length (Blens) Learning.**    The second component utilizes the inferred topology $\tau$ and genomic embeddings $E$ from DNABERT2 to generate and adjust branch lengths.

**Step1:  Node Feature via Linear-Time Dual-Pass Traversal.**    We utilize a linear time $\mathcal{O}(n)$ dual-pass traversal method, combining postorder (bottom-up) and preorder (top-down) strategies, guaranteeing each node is processed only once.

Postorder Traversal (bottom-up) aggregates information from leaf nodes toward the root node.
$$c_i = \max(1, K - \sum_{j \in \mathrm{ch}(i)} c_j)^{-1}, \tag{1}$$

where $c_i$ is the scaling factor for node $i$, influenced by the node's connectivity, and $K = 3$ due to binary tree properties.
$$f_i = c_i \cdot f_j + e_i, \tag{2}$$

where $f_i$ incorporates contributions from children nodes $f_j$ and its own initial feature $e_i$ from a pre-trained genome model. Initially, $c_i = 0, f_i = e_i$.

Preorder Traversal (top-down) propagates information from the root node toward the leaf nodes, enhancing each node's feature $x_i$:
$$x_i = c_i \cdot e_{p[i]} + f_i, \tag{3}$$

where $e_{p[i]}$ is the feature of node $i$'s parent.

**Step2:  Feature Enhancement with Dynamic Graph Convolution (DGCNN)**   : We use the edge convolutional layer of DGCNN [35] to enhance node features. This approach captures both local interactions (between neighboring nodes) and global structural dependencies (reflecting the entire tree structure) within the phylogenetic tree. For layer $L$, inputs $\{x_i^L \in \mathbb{R}^F\}_{i=1}^N$ and outputs $\{x_i^{L+1} \in \mathbb{R}^{F'}\}_{i=1}^N$ have feature dimensions $F = 768$ and $F' = 100$. The transformations are:
$$x_i^{L+1} = \mathrm{DGCNN}(x_i^L), \quad m_{ij} = h(x_i^L, x_j^L),$$

where $h : \mathbb{R}^F \times \mathbb{R}^F \to \mathbb{R}^{F'}$ is an MLP-shared asymmetric edge function.

Topology-Invariance Property. Despite the non-uniqueness of 2D coordinates due to infinite possible translational rotations, the distances between species remain constant. We define this as topological invariance: $d_{ij}^2 = \|z_i - z_j\|_2$, where $z_i$ and $z_j$ are coordinates in the hidden space, maintaining accurate topological relationships. The function $h(x_i, x_j)$ incorporates global and local features through: $h(x_i, x_j) = h(x_i, x_i - x_j, d_{ij}^2)$.

Aggregation of Features. The node features are aggregated using a MAX operation across all edges:

$$x_i^{L+1} = \sum_{k=1}^{K} m_{ij}^L = m_{i1}^L \oplus \ldots \oplus m_{iK}^L, \tag{4}$$

where $\oplus$ denotes the MAX aggregation function, focusing on the most significant features.

**Step3: Reparametrization of Branch Length.** Node features $x_i^L$ obtained from edge convolution layers are further fed into the M network, parameterizing mean and log-variance for branch lengths: $\mu, \log(\sigma^2) = \text{MLP}_2(h_{ij}^{(1)})$. Concretely, the M network is defined as:

$$h_i^{(1)} = \text{MLP}_1(x_i^L), \quad h_{ij}^{(1)} = \text{MAX}\{h_j^{(1)} : j \in \mathcal{N}(i)\} \cup h_i^{(1)}, \tag{5}$$

where $h_i$ are node features processed through an additional $\text{MLP}_1$ to capture inter-node interactions. Branch lengths are updated by reparametrization $b = \exp(\mu + \exp(\sigma) * \text{rvs})$, where $\text{rvs} \sim \mathcal{N}(0, I)$, ensuring differentiability and capturing the probabilistic nature of estimates. Throughout, the prior $P(B_\tau)$ is assumed exponential, while the posterior $Q(B_\tau)$, learned via a graph neural network, reflects inferred tree topologies and node characteristics.

### 3.1 Scoring Function

To address the convergence challenges often associated with the ELBO in VAE models, we incorporate a scoring function $S$, implemented via an MLP network. This function assesses each leaf node in the latent space $z$ and provides additional gradient information, facilitating more efficient learning and convergence.

During training, $S$ and ELBO form a joint optimization objective, optimizing gradient directions to improve overall performance. Fig. 3 compares the convergence behaviors and stability of $S$ and ELBO throughout the training process. The horizontal axis represents the training steps, and the vertical axis represents the two metric values. The **closer** the $S$ curve is to the ELBO curve, the more it proves

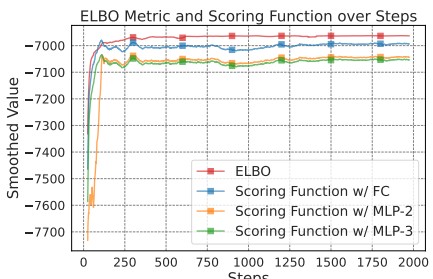

Figure 3: Comparison of ELBO and Scoring Function over Training Steps on DS1. **Closer curves** mean **better**.

that $S$ can effectively evaluate the model performance and maintain a consistent optimization trend with ELBO. Different configurations of $S$, including those with Fully Connected layers (w/ FC), with two layers MLP (w/ MLP-2), and with three layers MLP (w/ MLP-3), demonstrate similar trends, closely following the ELBO curve. After an initial period of rapid change, all metrics stabilize and exhibit minor fluctuations, demonstrating robustness in convergence. What's more, the number of layers in MLP has less impact on performance.

### 3.2 Learning Objectives

As discussed in Appendix Background A, our primary goal is to maximize the expected marginal likelihood of the observed species sequence $Y$ via $\max \log p(Y|(\tau(z), B_\tau))$. The posterior distribution as $p(\tau(z), B_\tau|Y)$ is difficult to infer directly, so we utilize variational inference to approximate it as $q(\tau(z), B_\tau|Y)$. The detailed deviation is in Appendix D.2.

To minimize the KL divergence between the true prior and the approximate posterior distributions, we start from the joint probability distribution: $p(Y, \tau(z), B_\tau) = p(Y|\tau(z), B_\tau)p(B_\tau|\tau(z))p(\tau(z))$, where $p(Y|\tau(z), B_\tau)$ represents the conditional probability of the observed data $Y$. We assume the tree topology $\tau(z)$ and the branch lengths $B_\tau$ are conditionally independent.

We introduce a variational distribution $q(\tau(z), B_\tau) = q(B_\tau|\tau(z))q(\tau(z))$ to approximate the true posterior $p(\tau(z), B_\tau|Y)$. The ELBO loss can initially be written as:

$$\mathcal{L}(Q) = \mathbb{E}_q[\log p(Y, \tau(z), B_\tau)] - \mathbb{E}_q[\log q(\tau(z), B_\tau)]. \tag{6}$$

To improve training variance and gradient stability, we introduce a regularization term $R(z|\tau(z^*))$ and reformulate the ELBO loss as:

$$\mathcal{L}(Q, R) = \mathbb{E}_{Q(z)}[\mathbb{E}_{Q(B_\tau|\tau(z))}[\log \frac{p(Y, B_\tau|\tau(z))p(\tau(z))R(z|\tau(z))}{Q(B_\tau|\tau(z))Q(z^*)}]]. \tag{7}$$

For better performance and reduced variance, we adopt a multi-sample approach[23]:

$$\mathcal{L}_{\text{multi-sample}}(Q, R) = \frac{1}{K} \sum_{k=1}^{K} \log \frac{p(Y, B_\tau^k \mid \tau(z^k))p(\tau(z^k))R(z^k \mid \tau(z^{*k}))}{Q(B_\tau^k \mid \tau(z^k))Q(z^{*k})}, \tag{8}$$

where $z^{*k}$ and $B_\tau^k$ are samples from the variational distributions $Q(z^*)$ and $Q(B_\tau|\tau(z))$, respectively. $K$ represents the number of Monte Carlo samples from the variational distribution to compute the empirical average, providing a more robust approximation of the ELBO. Typically, $K$ is chosen to be moderate to balance between computational efficiency and estimation accuracy [18]. And the multi-sample ELBO reduces to the standard ELBO formula when $K$=1.

The total loss $\mathcal{L}_{total}$ is then defined as:

$$\mathcal{L}_{total} = -\mathcal{L}_{\text{multi-sample}}(Q, R) + \mathcal{L}(S) + \mathcal{L}_{KL}, \tag{9}$$

where $\mathcal{L}_{KL} = -\text{KL}\{q_\phi(\tau(z)|y)||p_\theta(\tau(z))\} - \text{KL}\{q_\phi(B_\tau|y)||p_\theta(B_\tau)\}$, minimizing the KL divergence, and $\mathcal{L}(S) = -\sum_{i=1}^{N} f(z_i)$ accounts for the scoring function loss of the embeddings.

### 3.3 End-to-End Learning via Stochastic Gradient Descent

We derive the gradients of model parameters $\theta$ as follows:

$$\nabla_\theta \mathcal{L} = \mathbb{E}_{Q_\theta(z^*)}[\nabla \log Q_\theta(z^*) \mathbb{E}_{Q(B_\tau|\tau(z))}[\log \frac{P(Y, B_\tau|\tau(z))}{Q(B_\tau|\tau(z))}] +$$
$$\log P(\tau(z))R(z|\tau(z^*))] + \nabla H[Q_\theta(z^*)], \tag{10}$$

where $H[Q_\theta(z^*)]$ is the entropy of $Q_\theta(z^*)$. The derivation of the model parameters $\phi$:

$$\nabla_\phi \mathcal{L} = \nabla_\phi \mathbb{E}_{Q_\theta(z^*)}[\mathbb{E}_{Q_\phi(B_\tau|\tau(z))} \log \frac{P(Y, B_\tau \mid \tau(z))}{Q(B_\tau \mid \tau(z)))}]. \tag{11}$$

The derivation of the model parameters $\psi$:

$$\nabla_\psi \mathcal{L} = \mathbb{E}_{Q_\theta(z^*)}[\nabla_\psi \log (R_\psi(z|\tau(z^*)))]. \tag{12}$$

## 4 Experiments

In this section, we conduct experiments to demonstrate the effectiveness of our proposed PhyloGen. We aim to answer seven research questions as follows:

**RQ1:** How effective is PhyloGen in generating tree structures under the benchmark datasets?
**RQ2:** How diverse are the tree topologies generated by PhyloGen?
**RQ3:** How consistent is the PhyloGen-generated tree structure compared to the MrBayes method?
**RQ4:** How robust is PhyloGen to species sequences?
**RQ5:** How does each PhyloGen's module affect its performance?
**RQ6:** What evolutionary relationships between species does PhyloGen learn?
**RQ7:** How do key hyper-parameters affect PhyloGen's performance?

### 4.1 Experiment Setup

**Tasks and Datasets.** We evaluate the performance of PhyloGen on the Variational Bayesian Phylogenetic Inference task with Evidence Lower Bound (ELBO) and Marginal Log Likelihood (MLL) as metrics on eight benchmark datasets (see Appendix C).

**Baselines.** We compared PhyloGen against three categories of methods: MCMC-based methods like MrBayes and SBN. Structure Representation methods, including VBPI and VBPI-GNN, use pre-generated tree topologies in training that have the potential to achieve high likelihoods, and thus, their training and inference are restricted to a small space of tree topologies and thus are not directly comparable. Structure Generation methods, which are PhyloTree Structure Generation tasks that perform approximate Bayesian inference without pre-selecting topologies. For additional experimental details, including training details, baselines, architectures, and hyperparameters, the interested reader is referred to Appendix E.

## 4.2 Performance evaluation across eight benchmark datasets (RQ1)

Table 1: Comparison of the MLL (↑) with different approaches in eight benchmark datasets. VBPI and VBPI-GNN use pre-generated tree topologies in training and thus **are not directly comparable**. **Boldface** for the highest result, underline for the second highest from traditional methods, and underline for the second highest from tree structure generation methods.

| Methods | Dataset
#Taxa (N) | DS1
27 | DS2
29 | DS3
36 | DS4
41 | DS5
50 | DS6
50 | DS7
59 | DS8
64 |
|---|---|---|---|---|---|---|---|---|---|
| MCMC-based | MrBayes | -7108.42
(0.18) | -26367.57
(0.48) | -33735.44
(0.50) | -13330.44
(0.54) | -8214.51
(0.28) | -6724.07
(0.86) | -37332.76
(2.42) | -8649.88
(1.75) |
| | SBN | -7108.41
(0.15) | -26367.71
(0.08) | -33735.09
(0.09) | -13329.94
(0.20) | -8214.62
(0.40) | -6724.37
(0.43) | -37331.97
(0.28) | -8650.64
(0.50) |
| Structure Representation | VBPI | -7108.42
(0.10) | -26367.72
(0.12) | -33735.10
(0.11) | -13329.94
(0.31) | -8214.61
(0.67) | -6724.34
(0.68) | -37332.03
(0.43) | -8650.63
(0.55) |
| | VBPI-GNN | -7108.41
(0.14) | -26367.73
(0.07) | -33735.12
(0.09) | -13329.94
(0.19) | -8214.64
(0.38) | -6724.37
(0.40) | -37332.04
(0.12) | -8650.65
(0.45) |
| Structure Generation | ARTree | -7108.41
(0.19) | -26367.71
(0.07) | -33735.09
(0.09) | -13329.94
(0.17) | -8214.59
(0.34) | -6724.37
(0.46) | -37331.95
(0.27) | -8650.61
(0.48) |
| | phi-CSMC | -7290.36
(7.23) | -30568.49
(31.34) | -33798.06
(6.62) | -13582.24
(35.08) | -8367.51
(8.87) | -7013.83
(16.99) | NA | -9209.18
(18.03) |
| | GeoPhy | -7111.55
(0.07) | -26379.48
(11.60) | -33757.79
(8.07) | -133342.71
(1.61) | -8240.87
(9.80) | -6735.14
(2.64) | -37377.86
(29.48) | -8663.51
(6.85) |
| | GeoPhy LOO(3)+ | -7116.09
(10.67) | -26368.54
(0.12) | -33735.85
(0.12) | -13337.42
(1.32) | -8233.89
(6.63) | -6735.9
(1.13) | -37358.96
(13.06) | -8660.48
(0.78) |
| | PhyloGFN | -7108.95
(0.06) | -26368.9
(0.28) | -33735.6
(0.35) | -13331.83
(0.19) | -8215.15
(0.20) | -6730.68
(0.54) | -37359.96
(1.14) | -8654.76
(0.19) |
| | **Ours** | **-6910.02**
(0.07) | **-26257.09**
(0.06) | **-33481.57**
(0.10) | **-13063.15**
(1.34) | **-7928.4**
(0.23) | **-6330.21**
(0.31) | **-36838.42**
(12.03) | **-8171.04**
(0.96) |

Table 2: Comparison of ELBO (↑) on eight datasets. GeoPhy is not provided in the original paper and is tested by us. **Boldface** for the highest result, underline for the second highest result.

| Methods | Dataset
#Taxa (N) | DS1
27 | DS2
29 | DS3
36 | DS4
41 | DS5
50 | DS6
50 | DS7
59 | DS8
64 |
|---|---|---|---|---|---|---|---|---|---|
| MCMC-based | SBN | -7110.24
(0.03) | -26368.88
(0.03) | -33736.22
(0.02) | -13331.83
(0.02) | -8217.80
(0.04) | -6728.65
(0.04) | -37334.85
(0.03) | -8655.05
(0.04) |
| Structure Generation | GeoPhy | -7116.67
(1.71) | -26434.84
(0.10) | -33766.72
(0.15) | -13389.36
(3.45) | -8220.91
(2.64) | -6769.41
(3.25) | -37882.96
(1.97) | -8654.39
(0.97) |
| | ARTree | -7110.09
(0.04) | -26368.78
(0.07) | -33735.25
(0.08) | -13330.27
(0.05) | -8215.34
(0.04) | -6725.33
(0.06) | -37332.54
(0.13) | -8651.73
(0.05) |
| | **Ours** | **-7005.98**
(0.06) | **-26362.75**
(0.12) | **-33430.94**
(0.34) | **-13113.03**
(3.67) | **-8053.23**
(2.58) | **-6324.9**
(1.26) | **-36838.42**
(1.97) | **-8409.06**
(1.07) |

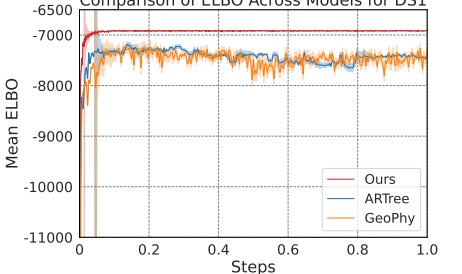
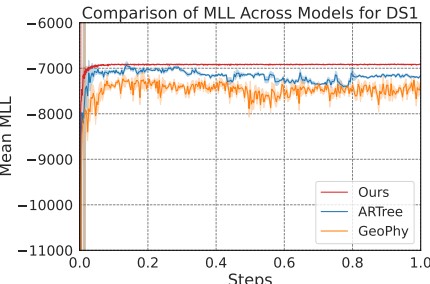

Figure 4: Comparison of ELBO and MLL Metrics for DS1 Dataset with Different Baselines.

We compare the MLL and ELBO metrics of various phylogenetic inference methods on eight benchmark datasets. Tab. 1 and Tab. 2 show that methods that utilize Tree Structure Generation have wider applicability than Structure Representation methods, which are restricted to limited pre-generated topologies. Our method PhyloGen outperforms other methods, achieving the highest MLL

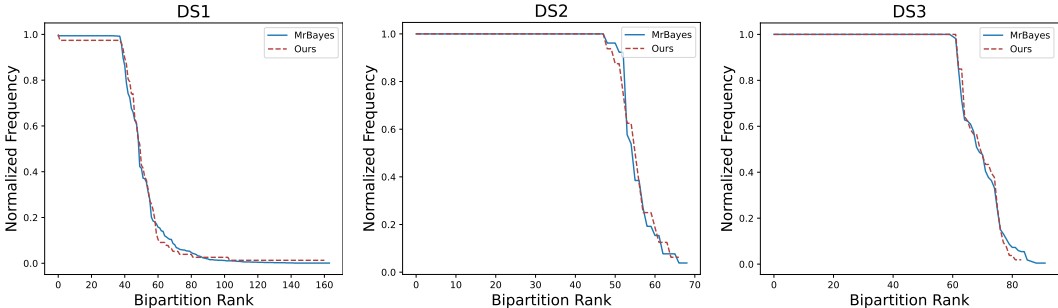

Figure 5: Comparative Bipartition Frequency Distribution in Tree Topologies for DS1, DS2, and DS3 datasets. **The closer the two curves are, the better,** which suggests that our method is highly consistent with the gold standard MrBayes approach.

and ELBO values on all datasets. The left plot of Fig. 4 illustrates our model's high stability and rapid convergence in ELBO metrics on DS1, which significantly outperforms the competition. ARTree performance improves in the later stages but exhibits large fluctuation. GeoPhy performs the worst, with consistently the lowest and more fluctuating ELBO values. The right plot of Fig. 4 demonstrates our model's advantages in the MLL metric, where it rapidly achieves and maintains high-performance levels. In contrast, ARTree and GeoPhy have lower MLL values, especially GeoPhy, which has the weakest performance throughout.

## 4.3 Tree Topological Diversity Analysis (RQ2)

To evaluate the topological diversity of trees generated by PhyloGen on DS1, we use three metrics: Simpson's Diversity Index [5], Top Frequency, and Top 95% Frequency, as detailed in Tab. 3. A higher Diversity Index, which approaches 1, suggests broad diversity among generated tree topolo-

Table 3: Diversity of tree topologies.

| Statistics | MrBayes | GeoPhy | Ours |
|---|---|---|---|
| Diversity Index (↑) | 0.87 | 0.36 | **0.89** |
| Top Frequency (↓) | 0.27 | 0.80 | **0.008** |
| Top 95% Frequency (↑) | 42 | 11 | **149** |

gies. The lower Top Frequency suggests a balanced distribution, preventing single tree structures from being overly dominant. Furthermore, the presence of 149 distinct topologies within the Top 95% Frequency underscores PhyloGen's ability to generate a diverse range of topologies.

## 4.4 Bipartition Frequency Distribution (RQ3)

Fig. 5 shows the bipartition frequency distributions of trees inferred by PhyloGen for datasets DS1, DS2, and DS3. The horizontal axis indicates the ranking of the bipartitions in the tree topology, and the vertical axis indicates the normalized frequency of occurrence of the corresponding bipartitions. The **similarity** of our method's **curves** to those of MrBayes underscores its accuracy, demonstrating that PhyloGen consistently captures evolutionary patterns with reliability comparable to the gold standard. This indicates a robust validation of PhyloGen's phylogenetic inference capabilities. More detailed information is provided in Appendix E.3.

Table 4: Model Robustness Assessment. Performance is evaluated by ELBO (↑) and MLL (↑). (Δ) represents the absolute difference change after node additions and deletions, with positive values indicating improved performance and negative values indicating a decline. Time records the total computation duration.

| | Metric | PhyloGFN | GeoPhy | GeoPhy LOO(3)+ | Ours w/o KL | Ours w/o S | Ours |
|---|---|---|---|---|---|---|---|
| setting1 | **ELBO (Δ)** | NA | -7721.82 (-100) | -7729.28 (-107) | -6725.49 (+12) | -6713.01 (+15) | **-6711.47 (+14)** |
| | **MLL (Δ)** | -6705.55 (-93) | -7440.38 (-198) | -7599.85 (-357) | -6564.51 (+18) | -6547.32 (+20) | **-6542.75 (+22)** |
| | **Time** | 18h28min | 6h16min | 15h23min | 6h43min | 7h42min | **6h32min** |
| setting2 | **ELBO (Δ)** | NA | -11802.07 (-98) | -11676.29 (-76) | -10678.24 (+3) | -10655.02 (+2) | **-10674.28 (+4)** |
| | **MLL (Δ)** | -12565.76 (-233) | -11763.40 (-131) | -11630.16 (-98) | -10422.12 (+6) | -10654.53 (+12) | **-10432.71 (+5)** |
| | **Time** | 24h35min | 18h13min | 12h24min | 7h54min | 8h6min | **6h37min** |

## 4.5 Robustness Assessment (RQ4)

To assess our model's robustness, we test its adaptability to data changes by modifying the number of nodes in the DS1 dataset, which initially contained 27 species sequences. Specifically, we conduct two experiments: Setting 1: randomly deleting 4 nodes to simulate the impact of data incompleteness and potential information loss, and Setting 2: randomly adding 4 nodes to simulate an increase in data size. As shown in Tab. 4, our model and its variants exhibit significant stability and adaptability under both cases. Changes in ELBO and MLL metrics are represented by Δ values, where positive changes indicate improved performance and negative changes indicate decreased performance. Specifically, smaller positive increases after node deletion (Setting 1) and node addition (Setting 2) emphasize the model's ability to adapt to changes in data structure effectively. In contrast, larger negative decreases highlight challenges when adjusting to increased data complexity. Furthermore, our model exhibits considerable computational efficiency, outperforming baselines in runtime, a critical advantage for handling the complexities and variabilities of bioinformatics datasets.

## 4.6 Ablation Study (RQ5)

Tab. 5 shows the performance of our model compared with the removal of the KL loss and the Scoring Function S, respectively. Ours performs best on ELBO and MLL metrics, with a significant decrease in performance after the removal of the KL loss, suggesting that the KL loss plays a key role in regularizing the model and avoiding overfitting. While removing the S module had a large

Table 5: Ablation Study of PhyloGen.

| Methods | ELBO (↑) | MLL (↑) |
|---|---|---|
| Ours | **-7005.98** | **-6910.02** |
| Ours w/o KL | -7017.57 | -6917.34 |
| Ours w/o S | -7011.94 | -6919.39 |

impact on MLL, the ELBO impact was relatively small, indicating that the impact of the S module is more complex and may be related to specific feature extraction functions. Our future work will explore further enhancements to the S module and investigate other regularization techniques to refine the model's performance.

Fig. 6 shows that our method (PGen) achieves the highest MLL value, indicating optimal fit and stability throughout the training process. Adjustments to the model structure, particularly without layer normalisation (PGen w/o LN) and reducing the hidden dimensions (PGen-Hid=64), result in a lower MLL, but convergence remains stable. These results highlight the model's sensitivity to hyperparameters and affirm its robustness under different configurations. Replacing our designed distance matrix D with both Euclidean (PGen-Euclidean) and cosine (PGen-Cosine) distance matrices would greatly affect the effective-

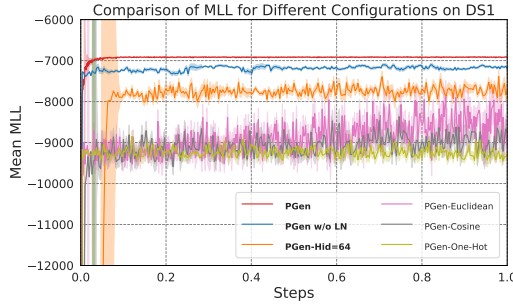

Figure 6: Ablation Study on DS1 Dataset.

ness of MLL. These methods do not capture complex evolutionary relationships as effectively as distance matrices derived from the potential space $z$. The model with one-hot encoding as the leaf node representation (PGen-One-Hot) has the lowest MLL, which indicates the importance of our feature extraction module.

## 4.7 Case Study of PhyloTree Structure (RQ6)

Fig. 7 shows a phylogenetic tree constructed from DS1 dataset, where each leaf node represents a specific species, and the text next to the node indicates the species name. The branch lengths reflect the genetic distances, with shorter branches indicating recent evolutionary history and longer branches indicating greater genetic differences. The phylogenetic tree shown in Fig. 7 places Siren intermedia and Trachemys scripta, both aquatic organisms, on adjacent branches, reflecting our model could capture their adaptive evolutionary information to aquatic environments. Meanwhile, the reptilian species Heterodon platyrhinos and Trachemys scripta are also on neighbouring branches, suggesting a relatively recent com-

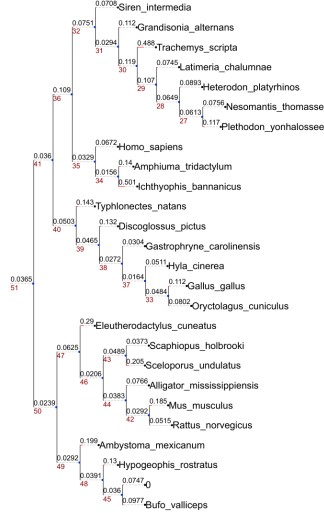

Figure 7: Plot of PhyloTrees.

mon ancestor compared to other amphibian and reptilian species. Notably, our method does not require padding the sequences to a unified length, which effectively reflects actual sequence variation. For a detailed species sequence and topology comparison, please refer to Appendix E.6.

## 5  Conclusion

**Contributions**   In this study, we introduced PhyloGen, a novel approach leveraging pre-trained genomic language models to enhance phylogenetic tree inference through graph structure generation. By addressing the limitations of traditional MCMC and existing VI methods, PhyloGen jointly optimizes tree topology and branch lengths without relying on evolutionary models or equal-length sequence constraints. PhyloGen views phylogenetic tree inference as a conditionally constrained **tree structure generation** problem, jointly optimizing tree topology and branch lengths through three core modules: (i) Feature Extraction, (ii) PhyloTree Construction, and (iii) PhyloTree Structure Modeling. These modules map species sequences into a continuous geometric space, refine the tree topology and branch lengths, and maintain topological invariance. Our method demonstrated superior performance and robustness across multiple real-world datasets, providing deeper insights into phylogenetic relationships.

**Limitations and Future Works**   While our model demonstrates outstanding performance on standard benchmarks, it may benefit from using more expressive distributions or incorporating prior constraints to better capture complex dependencies and interactions in the latent space. Additionally, although the Neighbor-Joining algorithm is effective for iterative tree construction, it is computationally intensive. We are exploring efficient data structures and parallel processing techniques to address this bottleneck. Furthermore, our model has primarily been applied to genomic data, and further research is needed to extend its applicability to diverse biological data, such as protein and single-cell data.

## Acknowledgement

This work was supported by National Science and Technology Major Project (No. 2022ZD0115101), National Natural Science Foundation of China Project (No. U21A20427), Project (No. WU2022A009) from the Center of Synthetic Biology and Integrated Bioengineering of Westlake University and Integrated Bioengineering of Westlake University and Project (No. WU2023C019) from the Westlake University Industries of the Future Research Funding. We thank the AI Station of Westlake University for the support of GPUs.

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

# A  Background

## A.1  Graph Structure Generation (GSG)

Let $\mathcal{G} = (A, X)$ denote a graph, where $A \in \mathbb{R}^{N \times N}$ is the adjacency matrix and $X \in \mathbb{R}^{N \times F}$ is the node feature matrix with $x_i \in \mathbb{R}^F$ being the embedding of node $v_i$. Given a feature matrix $X$, the target of GSG is to directly learn a graph structure $A^*$ by jointly optimizing the graph structure and Graph Neural Networks (GNN) [19, 1, 3].

**Feature Extraction** To better model the similarity of node pairs, a feature extractor is usually needed to map node features from a high-dimensional input space to a low-dimensional embedding space [40].

**Graph Construction** A similarity metric function is generally used to compute the similarity between embedding pairs as edge weights. There are several ways to construct a sparse adjacency matrix from a fully connected similarity matrix. For instance, we can create a graph that selects only connected node pairs whose similarity exceeds some predetermined threshold. In addition, we can connect the k nearest nodes to one node, thus constructing a k-nearest neighbor (kNN) [28].

**Graph Structure modeling** The core of GSG is the structure learner, which models edge connectivity to refine the preliminary graph [12]. Metric-based and neural network-based approaches are generally employed to learn edge connectivity through parameterized networks to receive node representations and regenerate the adjacency matrix that optimizes the graph structure, which is able to reveal the real connectivity relationship between nodes better and can be widely used in various downstream tasks [26].

## A.2  Variational Inference (VI)

Variational Autoencoders (VAE) [15] is a deep generative models that learn the distribution of input data by encoding it into a latent space. In this process, the encoder maps each input $x$ to a latent space defined by parameters: mean $\mu$ and variance $\sigma$. Latent variables $z$ are then sampled from this distribution for data generation.

VI is employed within VAE to handle the computational challenges of estimating marginal likelihoods of observed data. This involves computing the log of the marginal likelihood:

$$\max_{\theta} \log p_{\theta}(X) = \sum_{i=1}^{N} \log \int_{Z} p_{\theta}(X, Z) \mathrm{d}z \tag{13}$$

where $p_{\theta}(X, Z)$ represents the joint distribution of the observable data $x$ e.g. a Gaussian distribution, $\mathcal{N}(x|\mu, \sigma)$ and its latent encoding $Z$ under the model parameter $\theta$.

Since the direct estimation of marginal likelihoods is typically infeasible, VI introduces a variational distribution $q_{\phi}(z|x)$ to approximate the true posterior. The goal of VI is to maximize the Evidence Lower Bound (ELBO), formulated as:

$$\text{ELBO} = \mathbb{E}_{q_{\phi}(z|x)}[\log p_{\theta}(x|z)] - \text{KL}[q_{\phi}(z|x)||p(z)] \tag{14}$$

The first term is the reconstruction log-likelihood, $\log p_{\theta}(x|z)$ can be considered as a decoder, i.e., the log-likelihood between the reconstructed data and the original data given the potential representation. The second term, the KL divergence, quantifies the difference between the variational posterior $q_{\phi}(z|x)$ and the latent prior $p(z)$. Usually, VAE utilizes a reparameterization trick for gradient backpropagation through non-differentiable sampling operations. Once trained, VAEs can generate new data by directly sampling from the latent space and processing it through the decoder.

# B  Related Work

**MCMC-based methods**, such as MrBayes [29] and RevBayes [9], have been widely used for phylogenetic inference due to their ability to explore vast tree spaces. However, these methods are limited by the high-dimensional search space and the combinatorial explosion of tree topologies, which makes estimating posterior probabilities using simple sample relative frequencies (SRF) problematic [8]. The accuracy of these methods is compromised in unsampled tree spaces, and they

tend to be unstable when estimating low-probability trees, often requiring impractically large sample sizes to achieve reliable results.

**VI-based methods** offer a more efficient alternative to MCMC by leveraging approximate inference techniques. These methods can be categorized into two main approaches: structure representation and structure generation.

**Tree Representation Learning.** This approach focuses on extracting information from existing tree structures. Subsplit Bayesian Networks (SBN) [45]: Given a collection of trees, SBNs capture the relationships between existing subsplits, providing probabilistic representations of various tree shapes under given data. However, SBNs do not directly address branch lengths.

Variational Bayesian Phylogenetic Inference (VBPI) [46] and its variants VBPI-NF [43] and VBPI-GNN [44]: These methods introduce a two-pass approach to learn node representations, including branch lengths. VBPI employs variational approximations to handle branch lengths, allowing for joint modeling of the tree's latent space.

**Tree Structure Generation.** This approach aims to infer tree structures directly from sequence data. PhyloGFN [48] utilizes GFlowNet [10, 20, 21], a combination of VI and reinforcement learning. PhyloGFN constructs tree topologies by simplifying branch lengths into discrete intervals, requiring posterior data for accurate inference. VaiPhy [16] incorporates the SLANTIS sampling strategy [2] and basic biological models (e.g., Jukes-Cantor model) to estimate topology and branch lengths. ARTree [36] employs a graph autoregressive model to build detailed tree topologies. Branch length estimation is conducted independently through log-probabilities. Once the topology is determined, classical evolutionary models (e.g., Jukes-Cantor [24] or GTR (eneralized time reversible) [34]) are used to estimate branch lengths via maximum likelihood or Bayesian methods. GeoPhy [22] models tree topologies within a continuous geometric space, offering a different approach to the distribution of tree topologies. ARTree and GeoPhy differ primarily in their approach to modeling tree topologies. While ARTree uses a discrete autoregressive model, GeoPhy leverages continuous geometric representations, providing a more flexible framework for capturing the evolutionary relationships among species.

# C Datasets

Our model PhyloGen performs phylogenetic inference on biological sequence datasets of 27 to 64 species compiled in [17]. Notably, our approach does not require sequences to be of equal length, overcoming a common limitation in traditional phylogenetic analysis. In Tab. 6, we summarize the statistics of benchmark datasets.

Table 6: Statistics of the benchmark datasets from DS1 to DS8..

| Dataset | # Species | # Sites | Reference |
|---------|-----------|---------|-----------|
| DS1 | 27 | 1949 | [6] |
| DS2 | 29 | 2520 | [4] |
| DS3 | 36 | 1812 | [38] |
| DS4 | 41 | 1137 | [7] |
| DS5 | 50 | 378 | [17] |
| DS6 | 50 | 1133 | [47] |
| DS7 | 59 | 1824 | [39] |
| DS8 | 64 | 1008 | [30] |

# D Methods

## D.1 Scoring Function

To address the convergence challenges often associated with the ELBO in VAE models, we incorporate a scoring function $S$, implemented via an MLP network. This function assesses each leaf node in the latent space $z$ and provides additional gradient information, facilitating more efficient learning and

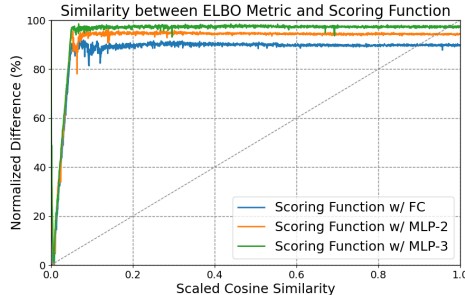

Figure 8: Analysis of the Cosine Similarities between Scoring Function and ELBO.

convergence. During training, $S$ and ELBO form a joint optimization objective, optimizing gradient directions to improve overall performance.

Fig. 3 compares the convergence behaviors and stability of $S$ and ELBO throughout the training process. The horizontal axis represents the training steps, and the vertical axis represents the two metric values. The **closer** the $S$ curve is to the ELBO curve, the more it proves that $S$ can effectively evaluate the model performance and maintain a consistent optimization trend with ELBO. Different configurations of $S$, including those with Fully Connected layers (w/ FC), with two layers MLP (w/ MLP-2), and with three layers MLP (w/ MLP-3), demonstrate similar trends, closely following the ELBO curve. After an initial period of rapid change, all metrics stabilize and exhibit minor fluctuations, demonstrating robustness in convergence. To further evaluate the performance of each $S$ configuration, we computed their scaled cosine similarity to ELBO. Scaled cosine similarity is calculated as: Scaled Cosine Similarity $= \frac{\mathbf{x} \cdot \mathbf{y}}{\|\mathbf{x}\|\|\mathbf{y}\|}$, where $\mathbf{x}$ and $\mathbf{y}$ are vectors of normalized performance metrics for $S$ and ELBO, respectively. As depicted in Figure 8, the horizontal axis measures the similarity ranging from 0 to 1, with values closer to 1 indicating a closer alignment with ELBO. The vertical axis displays the normalized difference percentage, highlighting how each configuration deviates from ELBO. All configurations maintain high similarity values, generally above 0.8, suggesting robust performance across different architectures. Given these results, the Fully Connected (w/ FC) configuration is selected for its closest to the ELBO curve and has better performance consistency. What's more, the number of layers in MLP has less impact on performance, as verified in Fig. 8 and Fig. 3.

### D.2 Gradient Derivation for Objective Function

It is important to emphasise that we no longer distinguish between $z^*$ and $z$ in the derivation for simplicity and uniformly use the latent variable $z$ to denote the topology.

In variational inference, we typically start from the joint probability distribution:

$$p(Y, \theta) = p(Y|\theta)p(\theta) \tag{15}$$

where $\theta$ could be any set of parameters or latent variables, and $Y$ is the observed data.

For a tree model, assume that $\theta$ includes the tree topology $\tau(z)$ and the branch lengths $B_\tau$. Thus, we have:

$$p(Y, \tau(z), B_\tau) = p(Y|\tau(z), B_\tau)p(\tau(z), B_\tau) \tag{16}$$

where $p(Y|\tau(z), B_\tau)$ represents the conditional probability of the observed data $Y$, given the tree structure and branch lengths, while $p(\tau(z), B_\tau)$ is the joint prior probability of the tree structure and branch lengths.

Further assuming that the tree topology $\tau(z)$ and the branch lengths $B_\tau$ are conditionally independent, the prior can be decomposed as:

$$p(\tau(z), B_\tau) = p(B_\tau|\tau(z))p(\tau(z)) \tag{17}$$

Combining the above formula 16 and 17, we can rewrite the joint probability as:

$$p(Y, \tau(z), B_\tau) = p(Y|\tau(z), B_\tau)p(B_\tau|\tau(z))p(\tau(z)) \tag{18}$$

In variational inference, we introduce a variational distribution $q(\tau(z), B_\tau)$ to approximate the true posterior distribution $p(\tau(z), B_\tau | Y)$, and often assume:

$$q(\tau(z), B_\tau) = q(B_\tau | \tau(z)) q(\tau(z)) \tag{19}$$

Thus, the ELBO can then be written as:

$$\mathcal{L}(Q) = \mathbb{E}_q \left[ \log p(Y, \tau(z), B_\tau) \right] - \mathbb{E}_q \left[ \log q(\tau(z), B_\tau) \right] \tag{20}$$

Inserting the joint distribution and variational distribution formulas, we obtain:

$$\mathcal{L}(Q) = \mathbb{E}_q \left[ \log p(Y|\tau(z), B_\tau) + \log p(B_\tau|\tau(z)) + \log p(\tau(z)) \right] - \mathbb{E}_q \left[ \log q(B_\tau|\tau(z)) + \log q(\tau(z)) \right] \tag{21}$$

This can be simplified by integrating the logarithmic terms:

$$\mathcal{L}(Q) = \mathbb{E}_q \left[ \log \frac{p(Y|\tau(z), B_\tau) p(B_\tau|\tau(z)) p(\tau(z))}{q(B_\tau|\tau(z)) q(\tau(z))} \right] \tag{22}$$

This step introduces the key P and Q ratio. We are essentially comparing the variational distribution $q$ and the conditional probability model $p$ for discrepancies.

The conditional distribution $C(z|\tau(z))$ is introduced to enhance the model's capacity for capturing complex dependencies within the data. This distribution is aimed at refining the approximation of latent variables given specific conditions. The revised ELBO, incorporating $C$, becomes:

$$\mathcal{L}(Q, R) = \mathbb{E}_{Q(z)Q(B_\tau|\tau(z))} \left[ \log \frac{p(Y, B_\tau|\tau(z)) p(\tau(z)) R(z|\tau(z))}{Q(B_\tau|\tau(z)) Q(z)} \right] \tag{23}$$

This formulation explicitly considers the influence of $C(z|\tau(z))$, reinforcing the variational framework's ability to more accurately model the posterior distributions conditioned on complex hierarchical data structures.

Since $\tau(z)$ and $B_\tau$ are sampled from their respective distributions and $B_\tau$ explicitly depends on the former:

$$\mathcal{L}(Q, R) = \mathbb{E}_{Q(z)} \left[ \mathbb{E}_{Q(B_\tau|\tau(z))} \left[ \log \frac{p(Y, B_\tau|\tau(z)) p(\tau(z)) R(z|\tau(z))}{Q(B_\tau|\tau(z)) Q(z)} \right] \right] \tag{24}$$

Begin the derivation of the model parameters $\theta$:

$$\nabla_\theta \mathcal{L} = \nabla_\theta \mathbb{E}_{Q_\theta(z)} \left[ \mathbb{E}_{Q(B_\tau|\tau(z))} \left[ \log \frac{p(Y, B_\tau|\tau(z)) p(\tau(z)) R(z|\tau(z))}{Q(B_\tau|\tau(z)) Q(z)} \right] \right] \tag{25}$$

$$\nabla_\theta \mathcal{L} = \nabla \left( \int Q_\theta(z) \mathbb{E}_{Q(B_\tau|\tau(z))} \left[ \log \frac{p(Y, B_\tau|\tau(z)) p(\tau(z)) R(z|\tau(z))}{Q(B_\tau|\tau(z)) Q(z)} \right] dz \right) \tag{26}$$

$$= \int Q_\theta(z) \nabla \log Q_\theta(z) \left( \mathbb{E}_{Q(B_\tau|\tau(z))} \left[ \log \frac{p(Y, B_\tau|\tau(z)) p(\tau(z)) R(z|\tau(z))}{Q(B_\tau|\tau(z)) Q(z)} \right] \right) dz \tag{27}$$

$$= \mathbb{E}_{Q_\theta(z)} \left[ \nabla \log Q_\theta(z) \left( \mathbb{E}_{Q(B_\tau|\tau(z))} \left[ \log \frac{p(Y, B_\tau|\tau(z)) p(\tau(z)) R(z|\tau(z))}{Q(B_\tau|\tau(z)) Q(z)} \right] \right) \right] \tag{28}$$

where $H[Q_\theta(z)] = -\mathbb{E}_{Q_\theta(z)}[\log Q_\theta(z)]$ is the differential entropy. The derivative of $H$ is $\nabla_\theta H[Q_\theta(z)] = -\nabla_\theta \mathbb{E}_{Q_\theta(z)}[\log Q_\theta(z)]$.

Then we use the chain rule to apply the derivative operation to the desired logarithmic term yields: $\nabla_\theta H[Q_\theta(z)] = -\mathbb{E}_{Q_\theta(z)}[\nabla_\theta \log Q_\theta(z)]$ and $\nabla_\theta \log Q_\theta(z) = \frac{\nabla_\theta Q_\theta(z)}{Q_\theta(z)}$, we rewrite the $\nabla_\theta \mathcal{L}$ as:

$$\nabla_\theta \mathcal{L} = \mathbb{E}_{Q_\theta(z)} [\nabla \log Q_\theta(z) \mathbb{E}_{Q(B_\tau|\tau(z))} [\log \frac{P(Y, B_\tau|\tau(z)) P(\tau(z)) R(z|\tau(z))}{Q(B_\tau|\tau(z)) Q(z)}]] \tag{29}$$

$$= \mathbb{E}_{Q_\theta(z)} [\nabla \log Q_\theta(z) \mathbb{E}_{Q(B_\tau|\tau(z))} [\log \frac{P(Y, B_\tau|\tau(z))}{Q(B_\tau|\tau(z))}] + \log P(\tau(z)) R(z|\tau(z))] + \nabla H[Q_\theta(z)] \tag{30}$$

Begin the derivation of the model parameters $\phi$:

$$\nabla_\phi \mathcal{L} = \nabla_\phi \mathbb{E}_{Q_\theta(z)} \left[ \mathbb{E}_{Q_\phi(B_\tau|\tau(z))} \left[ \log \frac{P(Y, B_\tau \mid \tau(z))}{Q(B_\tau \mid \tau(z)))} \right] \right] \tag{31}$$

We apply the reparameterization technique to express variables involving stochasticity as a combination of a deterministic transformation and a noise distribution that does not depend on the model parameters. With this transformation, the gradients of the model parameters can be passed directly through subsequent computations without being interrupted by the random sampling process. Next, we assume that $z$ can be expressed as a result of a deterministic transformation $h_\theta(\varepsilon_z)$.

Accordingly, $B_\tau$ is no longer sampled directly from a simple normal distribution but is instead obtained by transforming it through the function $h_\phi(\cdot)$, which combines the independent noise variable $\varepsilon_B$ (usually from a simple distribution such as the standard normal distribution) and $\tau(z)$ to generate $B_\tau$.

$$B_\tau = h_\phi(\varepsilon_B, \tau(z)), \varepsilon_B \sim p(\varepsilon) \tag{32}$$

Such treatment allows the model to take into account the fact that complex dependency structures into account, allowing the computation of gradients to be carried out in this reparameterized way, which is important for reflecting the actual complexity in biological data.

The expectation can be rewritten as:

$$\mathbb{E}_{Q_\phi(B_\tau|\tau(z))} = \mathbb{E}_{p(\varepsilon)}[\log \frac{P(Y, h_\phi(\varepsilon_B, \tau(z)) \mid \tau(z))}{Q(h_\phi(\varepsilon_B, \tau(z)) \mid \tau(z)))}] \tag{33}$$

Substituting this into the derivation formula gives:

$$\nabla_\phi \mathcal{L} = \nabla_\phi \mathbb{E}_{Q_\theta(z)}[\mathbb{E}_{Q_\phi(B_\tau|\tau(z))} \log \frac{P(Y, B_\tau \mid \tau(z))}{Q(B_\tau \mid \tau(z)))}] \tag{34}$$

$C_\psi(Z|\tau)$ Begin the derivation of the model parameters $\psi$:

$$\nabla_\psi \mathcal{L} = \mathbb{E}_{Q_\theta(z)}[\nabla_\psi \log (R_\psi(z|\tau(z))] \tag{35}$$

### D.3 Algorithm

PhyloGen's algorithm flow is as shown in Algorithm 1.

---

**Algorithm 1** Phylogenetic Tree Generation

---

1: **Input:** Genomic sequences $Y$, Model parameters
2: **Output:** Optimized phylogenetic tree $(\tau, B_\tau)$
3: **Preprocessing:** Encode $Y$ into genomic embeddings $E$ using DNABERT2.
4: **Initialize:** Generate initial latent variable $z^*$.
5: **Construct Tree:** Compute distance matrix $D$ from $z^*$ and use NJ algorithm.
6: **Optimize Tree Structure and Branch Lengths.**
7: **procedure** GRADIENTUPDATE($E$, $\tau(z^*)$, $\theta$, $\phi$, $\psi$)
8:     Initialize model parameters $\theta$, $\phi$, $\psi$
9:     **while** not converged **do**
10:         Sample $z^* \sim R_\psi(z^*|z)$ using the reparameterization trick.
11:         Sample $z \sim Q_\theta(z|Y)$ using the reparameterization trick.
12:         Compute the loss $\mathcal{L}_{total}$                   *(Eq. 9)*
13:         Compute gradients using backpropagation and Update parameters using Adam optimizer:
14:         $\theta \leftarrow \theta - \eta \nabla_\theta \mathcal{L}$            *(Eq. 10)*
15:         $\phi \leftarrow \phi - \eta \nabla_\phi \mathcal{L}$            *(Eq. 11)*
16:         $\psi \leftarrow \psi - \eta \nabla_\psi \mathcal{L}$            *(Eq. 12)*
17:     **end while**
18:     **return** Updated $\theta$, $\phi$, $\psi$
19: **end procedure**
20: **return** the generated phylogenetic tree $(\tau, B_\tau)$

---

# E Experiment

## E.1 Training Details

We focus on the most challenging aspect of the phylogenetic tree inference task: the joint learning of tree topologies and branch lengths. For this, we employ a uniform prior for the tree topology and an independent and identically distributed (i.i.d.) exponential prior (Exp(10)) for the branch lengths. We evaluate all methods across eight real datasets (DS1-8) frequently used to benchmark phylogenetic tree inference methods. These datasets include sequences from 27 to 64 eukaryote species, each comprising 378 to 2520 sites. Notably, our approach does not require sequences to be of equal length, thus overcoming a common limitation in traditional phylogenetic analysis. For our Monte Carlo simulations, we select $K = 2$ samples and apply an annealed unnormalized posterior during each $i$-th iteration, where $\lambda_n = \min\{1.0, 0.001 + i/H\}$ acts as the inverse temperature. This parameter starts at 0.001 and gradually increases to 1 over $H$ iterations, effectively simulating a cooling schedule commonly used in annealing algorithms, similar to the approach in [45], with an initial temperature of 0.001, which gradually decreases over 100,000 steps.

During the model training process, we utilize stochastic gradient descent to process a total of one million Monte Carlo samples, employing $K$ samples at each training step. The stepping-stone (SS) algorithm [37] in MrBayes is viewed as the gold-standard value. All models were implemented in Pytorch [27] with the Adam optimizer [14]. The MLL estimate is derived by sampling the importance of 1000 samples, with the larger mean value being better. The learning rate is initially set to 1e-4 and is reduced by a factor of 0.75 every 200,000 training steps. Momentum is set at 0.9 to prevent the optimization process from becoming trapped in local minima. Utilizing the StepLR scheduler, the current learning rate is multiplied by 0.75 every 200,000 steps to ensure steady progression, detailed in Tab. 7 and Tab. 8.

Table 7: Training Settings of PhyloGen.

| Training Configuration | |
| --- | --- |
| Optimizer | Adam optimizer |
| Learning rate | 1e-4 |
| Schedule | Step Learning Rate |
| Weight Decay | 0.0 |
| momentum | 0.9 |
| eta_min | 1e-6 |
| base_lr | 1e-4 |
| max_lr | 0.001 |
| scheduler.gamma | 0.75 |
| annealing init | 0.001 |
| annealing steps | 100,000 |

Table 8: Common Hyperparameters for PhyloGen.

| TopoNet | |
| --- | --- |
| Hidden Dim. | 256 |
| # Layer | 2 |
| Output Dim. | 4 |
| TreeEncoder | |
| Hidden Dim. | 256 |
| # Layer | 2 |
| TreeDecoder | |
| Hidden Dim. | 256 |
| # Layer | 2 |
| DGCNN | |
| # Layer | 2 |

## E.2 Baselines

We mainly compare PhyloGen with three types of methods, including two MCMC-based methods (i.e., MrBayes, SBN), two tree-structure representation learning methods (i.e., VBPI, VBPI-GNN), and five tree-structure generation methods. It should be noted that the results of all baseline methods are not included in the MLL tables, as some of the baseline methods are not provided with source code, and the results of the MLL metrics are not shown in the original paper.

## E.3 Bipartition Frequency Distribution (RQ3)

A bipartition frequency comparison plot is used in evolutionary analysis to show the bipartition frequencies observed in a sample of tree topologies. Each bipartition represents a node in the tree and defines the division of two sets of taxonomic units (e.g., species) on either side of that node. The frequency of these bipartitions in the posterior tree topology distribution reflects how often each topology appeared during the MCMC sampling process, which in turn reveals the confidence level of the different topological features.

We sample 1000 Monte Carlo trees from the posterior distribution $Q(\tau)$, calculate the frequency of each bipartition within these samples, and then compare these frequencies with the bipartition frequencies obtained using MrBayes. These frequencies are then compared with those obtained using the MrBayes method to visualize the consistency and differences between the two methods in terms of tree topology inference through graphs. Fig. 5 shows the bipartition frequency distributions of trees inferred by PhyloGen for the DS1, DS2, and DS3 datasets. The horizontal axis indicates the ranking of the bipartitions in the tree topology, and the vertical axis indicates the normalized frequency of occurrence of the corresponding bipartitions, reflecting the prevalence of various topological features observed during the MCMC sampling process. The results in the figure show that our method's variation curves are consistent with those of MrBayes, which indicates that our method has a high degree of consistency in tree topology inference and is able to capture underlying evolutionary patterns with a confidence level similar to the established gold standard MrBayes method.

### E.4 Ablation Study (RQ5)

Tab. 5 shows the performance of our model compared with the removal of the KL loss and the Scoring Function S, respectively. ours performs best on both the ELBO and MLL metrics, with a significant decrease in performance after the removal of the KL loss, suggesting that the KL loss plays a key role in regularizing the model and avoiding overfitting. While removing the S module had a large impact on MLL, the ELBO impact was relatively small, indicating that the impact of the S module is more complex and may be related to specific feature extraction functions. Our future work will explore further enhancements to the S module and investigate other regularization techniques to refine the model's performance.

Fig. 6 presents a comparative analysis of the MLL metric across six configurations of our model on the DS1 dataset. The standard configuration ('Ours') demonstrates superior performance with the highest MLL value, indicating optimal fit and stability throughout the training process. Variants without Layer Normalization or with a reduced hidden dimension show significant decreases in MLL metric. Furthermore, models using other distance functions for tree construction—specifically, Euclidean and Cosine distances—along with those using one-hot encoding for leaf node representation demonstrate lower MLL values. Notably, the one-hot approach works the worst, suggesting that both the method of computing the distance matrix and the choice of feature representation for leaf nodes critically influence the model's accuracy. Additionally, the distance matrix (calculated from the latent space $z$) that we specifically designed as an input to the NJ algorithm effectively captures key information about evolutionary relationships that a simple Euclidean or Cosine distance matrix would not be able to capture, resulting in MLL values lower than our standard configuration.

### E.5 Hyper-Parameter Analysis (RQ7)

Table 9: Hyperparameter Analysis of PhyloGen Performance in Various Parameter Configurations.

| Parameter Setting | Cfg. | ELBO (↑) | MLL (↑) |
|---|---|---|---|
| Output Dimension (emd) (hid=256) | 2 | -7111.95 | -6908.65 |
| | 3 | -7015.65 | -6914.82 |
| | 4 | -7012.20 | -6911.02 |
| | 8 | **-7005.98** | **-6910.02** |
| | 16 | -7023.67 | -6928.89 |
| Hidden Dimension (hid) (Layer Norm, emd=8) | 64 | -7116.75 | -6863.84 |
| | 128 | -7021.16 | -6917.80 |
| | 256 | **-7005.98** | **-6910.02** |
| | 512 | -7023.96 | -6915.18 |
| Hidden Dimension (No Layer Norm, emd=8) | 64 | -7123.71 | -6912.71 |
| | 128 | -7024.32 | -6919.42 |
| | 256 | -7020.07 | -6920.05 |
| | 512 | -7028.93 | -6921.01 |

The first part of Tab. 9 demonstrates the effect of different output dimensions on the model performance at a fixed hidden layer dimension (hd=256). It can be seen that the ELBO and MLL metrics

show different trends when the output dimension is increased from 2 to 16, especially when the output dimension is 8. The ELBO and MLL values are optimal, suggesting that higher output dimensions may contribute to the model performance, but there are some fluctuations. The second and third parts of Tab. 9 compare the effects of different hiding dimensions on model performance with and without using layer regularization. When using layer regularization, the model reaches optimal values for both ELBO and MLL at a hidden layer dimension of 256. The overall decrease in performance when layer regularization is not used shows that layer normalization plays an important role in model stability and performance.

## E.6 Visualizations

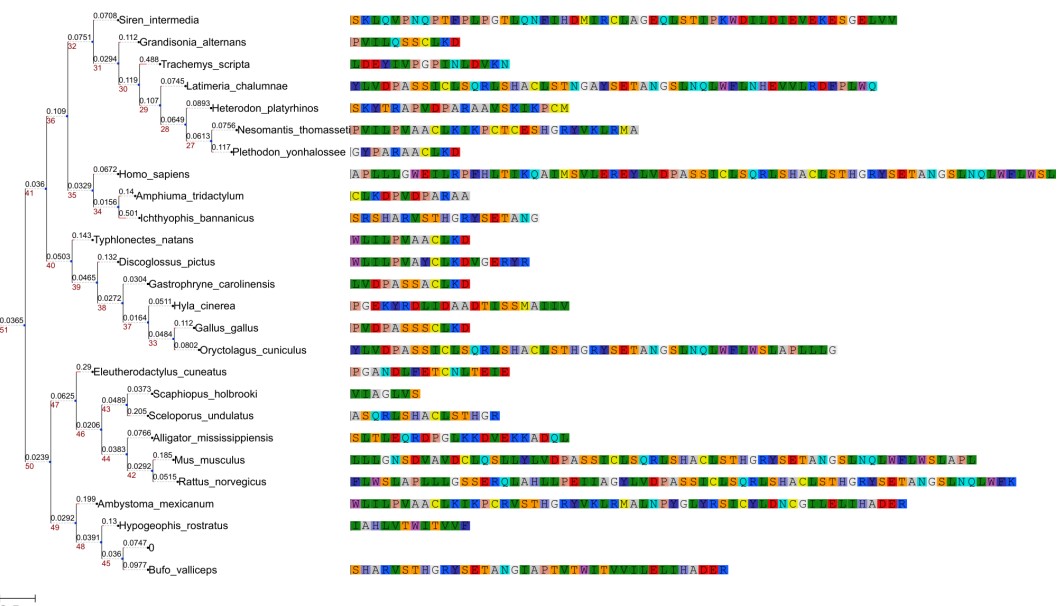

Figure 9: Visualization of phylogenetic trees. The left side shows a phylogenetic tree constructed from the sequences of the DS1 dataset. Each leaf node represents a specific species, and the text next to the node indicates the species name. On the right side, the colored sequences represent fragments of the species' protein sequences, with different colored blocks corresponding to different amino acid residues. It is worth noting that this method of sequence presentation does not require a uniform sequence length or padding procedure and can effectively reflect actual sequence variations. The scale bar in the lower left corner indicates the ratio between branch length and evolutionary distance. These sequence fragments visualize the key sequence features on which the construction of the phylogenetic tree depends.

## F Broader Impacts

We recognise the importance of addressing the societal impact of our work. Phylogenetic inference methods such as the one we propose have the potential to greatly improve our understanding of the evolution, origin, and transmission mechanisms of viruses and bacteria. Such an understanding could have far-reaching societal implications, especially in the fields of public health and disease control.

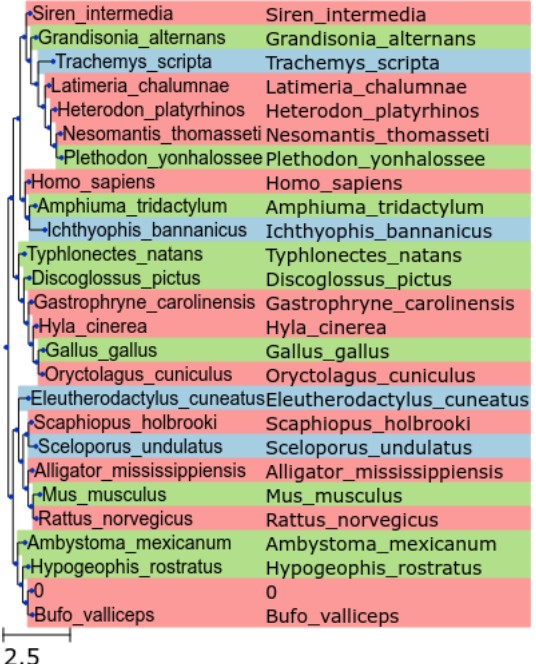

Figure 10: Enhanced visualization of phylogenetic relationships depicted through a coloured heatmap integrated with a phylogenetic tree. The evolutionary tree, structured using Newick format data, illustrates the hierarchical relationships among species. Nodes are distinctly coloured to represent the frequency of bipartition support derived from posterior probability analysis: high support (>0.2) is indicated with blue, medium support (>0.1) with green, and low support with red.

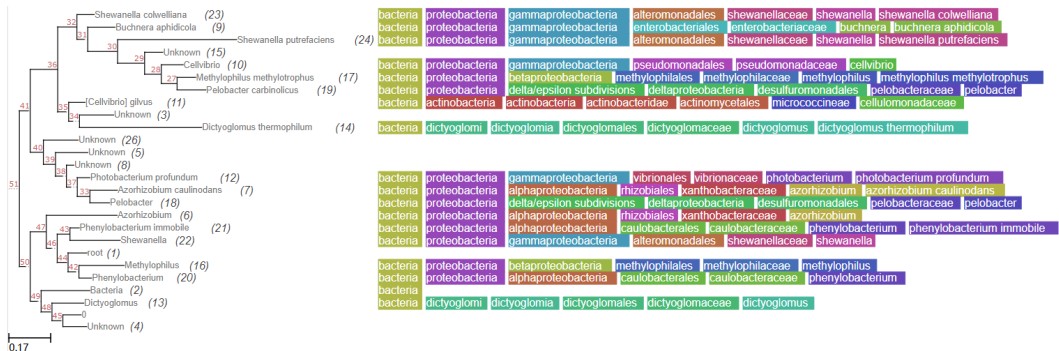

Figure 11: Phylogenetic tree visualization. This figure shows the phylogenetic relationships of bacterial taxa constructed based on comparative genomics analysis. Each node in the tree represents a species sample, and the labels on the right side show the detailed classification of the samples according to the taxonomic hierarchy, from broad taxonomic classes (e.g., "Bacteria") to more specific classes (e.g., "Genus" and "Species"). "species").

