# OpenReview forum: "PhyloGen: Language Model-Enhanced Phylogenetic Inference via Graph Structure Generation"
_NeurIPS.cc/2024/Conference — NeurIPS 2024 poster_

### Official Review · Reviewer_MKto · 2024-07-23

**Soundness:** 2
**Presentation:** 3
**Contribution:** 2
**Rating:** 7
**Confidence:** 4

**Summary:**

The authors introduce PhyloGen, a new method that uses a pre-trained genomic language model to generate phylogenetic trees without relying on evolutionary models or aligned sequences. PhyloGen treats phylogenetic inference as a conditionally constrained tree structure generation problem, jointly optimizing tree topology and branch lengths through three modules: Feature Extraction, PhyloTree Construction, and PhyloTree Structure Modeling. A Scoring Function guides the model towards stable gradient descent. The method achieves state-of-the-art performance when evaluated on benchmark datasets.

**Strengths:**

1. The framework is novel in that it leverages a pre-trained genomic language model  instead of relying on aligned sequences.
2. The method achieves state-of-the-art performance evaluated on real-world datasets.

**Weaknesses:**

1. Ablation Experiments Clarity

The ablation experiments are unclear. My main concern is the lack of clarity regarding why the proposed method significantly outperforms other methods. Specifically:

(1) Table 5: This table shows the average metrics across the eight datasets. However, Table 1 and Table 2 only shows the metrics for each dataset individually without any average metrics. It’s unclear what the performance gap is between the ablation model and the baseline methods when KL or S are removed.

(2) Figure 6: The ablation studies in this figure only show the metrics for the DS1 dataset. Please include the average metrics across all eight datasets.

To improve the clarity of the ablation experiments, I strongly suggest making the evaluation settings consistent in Table 1, Table 2,Table 5, and Figure 6. At the very least, include the average metrics across the eight datasets. Additionally, provide more explanations based on the complete ablation experiments to clarify why the proposed method significantly outperforms other methods.

2.  Claim on Aligned Sequences vs. Unaligned Sequences

The claim that unaligned sequences better reflect actual sequence variation than aligned sequences is questionable. Generally, aligned sequences with direct site-to-site comparisons are more effective in comparative sequence analysis. The authors need to provide more direct experimental evidence to support the advantage of using unaligned sequences.

3. Writing Issues

There are many writing issues. For instance,

a. line 141. "m_ij" -> "h"

b. In the caption of table1, "VBPI-GNNuse" -> "VBPI-GNN use"

c. line 213, "3.1 Experiment Setup"  ->  "4.1 Experiment Setup"

**Questions:**

1. Could you clarify whether the computation of the marginal log likelihood involves an evolutionary model like [1]? Does the evolutionary model utilize aligned sequences? If so, does this contradict the statement in the article that sequence alignment is not required?

2. I noticed that the author provides a comparison of run times. Could you provide details about the GPU used in the experiments, as well as the memory utilization?


[1] M. Zhou, Z. Yan, E. Layne, N. Malkin, D. Zhang, M. Jain, M. Blanchette, and Y. Bengio. Phy-logfn: Phylogenetic inference with generative flow networks. arXiv preprint arXiv:2310.08774, 2023.

**Limitations:**

I have no concerns on the potential societal impact of their work.

---

> ### Author Rebuttal · Authors · 2024-08-04
>
> **Weakness:**
>
> **W1: Table 5 Clarification:** Tab. 5 presents results for the DS1 dataset, not an average across eight datasets. The impact of removing KL or S has already been shown in this table.
>
> **Figure 6 Explanation:** Fig. 6 shows ablation results for the DS1 dataset, chosen for its representativeness, as many recent studies also use this dataset. This is a common practice in the field. Due to time constraints, we have added ablation results for some other datasets in the attached file. We can continue to add results if you feel it is necessary.
>
> **Performance Explanations:** The complete ablation experiments are shown in Tab. R1 indicates a significant performance decline when KL or S is removed, underscoring their importance in model regularization and overfitting prevention. Replacing our distance matrix with Euclidean or cosine distances significantly lowers MLL, indicating these conventional distance methods fail to capture complex evolutionary patterns. Using one-hot encoding exhibited the lowest MLL, highlighting the superiority of our feature extraction module and the pre-trained language model in representing complex biological data.
>
> **W2: Aligned vs. Unaligned Sequences**
>
> Thank you for your feedback. There might be some misunderstanding regarding the concept of "aligned sequences". In our manuscript, we refer to equal-length sequences, not multiple sequence alignments (MSA). This definition is consistent with recent methods like PhyloGFN and GeoPhy, which refer to "pre-aligned sequences" as uniform-length sequences, as seen in statements like "These datasets feature pre-aligned sequences" or "Let Y be a set of aligned sequences with length M from the species.". Our method analyzes raw sequences directly, avoiding biases introduced by alignment algorithms, thus preserving the original sequence diversity and more accurately reflecting sequence variation and evolutionary relationships.
>
> **W3: Writing Issues**
>
> Thank you for highlighting the writing issues. We will thoroughly review and improve the manuscript for clarity.
>
> **Questions:**
>
> **Q1: MLL:**
>
> Thank you for your question. Let me clarify these concepts:
>
> **PhyloGFN:**
>
> 1. **Evolutionary Model:** PhyloGFN uses an evolutionary model to compute the marginal log likelihood (mll) through trajectory sampling and importance sampling. This involves generating trajectory data, calculating the forward and backward path log probabilities, and computing the log scores for each generated tree.
> 2. **Aligned Sequences:** PhyloGFN does not rely on traditional multiple sequence alignment (MSA). Instead, it uses sequences of equal length, which do not require MSA for alignment.
>
> **Our Method:**
>
> 1. **Evolutionary Model:** Our mll computation does not involve PhyloGFN's evolutionary model. We use a mutation model based on mutation probabilities, not the trajectory and importance sampling methods used in PhyloGFN.
> 2. **Aligned Sequences:** Our method does not require either MSA or equal-length sequences. PhylGen directly analyzes raw sequence data, avoiding the computational overhead and potential errors introduced by sequence alignment. Thus, it more accurately reflects sequence variations and evolutionary relationships.
> 3. **Consistency with Claims:** Our method aligns with the statement that sequence alignment is unnecessary. By handling unaligned, variable-length raw sequences, PhylGen more accurately reflects sequence variation and evolutionary relationships. This makes our method distinct from others that rely on equal-length sequences, like PhyloGFN and GeoPhy, and consistent with the claims made in our article.
>
> **Q2: Memory Usage**
>
> Thank you for your question. We used an NVIDIA A100-SXM4-80GB GPU for our experiments. However, the choice of hardware is not crucial for running our training algorithm. Detailed memory usage can be found in the General Response and the Tab. R2.

---

> > ### Comment · Reviewer_MKto · 2024-08-12
> >
> > Thank you for your response. I raised my score. Please keep in mind to fix the typos and grammar errors in the final version of the manuscript.

---

> > > ### Author Response · Authors · 2024-08-12
> > > **Thanks for Raising the Score**
> > >
> > > Dear Reviewer MKto,
> > >
> > > We would like to express our sincere gratitude for your thorough review and for increasing your score after considering our response, which is very important to us. Based on your suggestions, we will keep on polishing our manuscript and add relevant tables to the camera-ready version. Thank you again for your valuable review and efforts in improving our work.
> > >
> > > Warm regards,
> > >
> > > Authors

---

> ### Author Response · Authors · 2024-08-10
>
> Dear Reviewer,
>
> We sincerely appreciate your efforts and valuable feedback. If you are satisfied with our responses and our improvements, please consider updating your score.
>
> If you need further clarification, please don't hesitate to contact us. We are grateful for your time and look forward to your response!

---

### Official Review · Reviewer_4WGb · 2024-07-26

**Soundness:** 2
**Presentation:** 2
**Contribution:** 2
**Rating:** 5
**Confidence:** 1

**Summary:**

The paper propose phylogenetic tree inference by modeling it as a problem of conditional-constrained tree structure generation. Its goal is to jointly generate and optimize the tree topology and branch lengths. By mapping species sequences into a continuous geometric space, PhyloGen performs end-to-end variational inference without limiting the possible topologies. To maintain the topology-invariance of phylogenetic trees, distance constraints are applied in the latent space to preserve translational and rotational invariance. They have shown the proposed model  is  effective  across eight real-world benchmark datasets.

**Strengths:**

The paper addresses very interesting problem and propose novel approches.

**Weaknesses:**

I am not sure if the formulation equation 7, adding the term R mathematically make sense, as orignal ELBO in equation 6, is the lower bound, now adding the term log of R which is negative, and trying to maximize equation 7 would result in minimizing R?  Also in equation 7, q(\tao (z)) seems missing, or is there a kl between R and Q from module B?
I think the paper presentation needs to be improved. The figure 2 is very helpful but the text  and some notations are a bit confusing.

**Questions:**

1. Regarding the statement, "Existing Variational Inference methods, which require pre-generated topologies and typi6 cally treat tree structures and branch lengths independently, may overlook critical sequence features, limiting their accuracy and flexibility", can you elaborate on what kind of features do you refer to that the method could overlook?

2. I am not sure fromt he distance matix to the tree topology, the  Neighbor-Joining (NJ) algorithm can be differentiated to be able to learm end to end?

3. I think the section C.1. Topology Learning is written somehow confusing. Especially from line 121 to line 124. Why the distribution  p(\tao(z)) is represented as the distribution of y conditioned on tree structure? same as the q(\tao(z), B_\tao) is not clear what distribution it represents.

4. I am not sure in this kind of tasks, if the ELBO is a good metric to reflect the inference power of the model.

5. The VAE model basically takes the embedding of the genomics sequences, and the initial topology of the tree \tao, then try to reconstruct the tree (without the leaf length)? I was thinking what happens if we remove the learning of the first component, make the phylo tree construction constant using directly the embeddings from LLM?

6. I am not sure if I understood how the scoring function is implemented, f(z), z is the embedding of the genomic sequence, f applied on the leaf nodes? or all the nodes, and why minimize it?

**Limitations:**

1. While PhyloGen outperforms many existing methods, it still requires substantial computational resources, particularly during the training phase. The complexity of jointly optimizing tree topology and branch lengths, along with maintaining topological invariance, sounds computationally heavy, for instance:
1.1  Monte Carlo simulations to sample from the posterior distributions of tree topologies and branch lengths.
1.2 The tree construction module computes distance matrices from latent variables and uses algorithms like Neighbor-Joining (NJ) to generate initial tree structures.
1.3 The utilization of  a pre-trained genome language model (DNABERT2) to transform DNA sequences into genomic embeddings,

2, Looking at the table 5, I am wondering how it works in inference time, would peformance drops when we have less or more specious than the one in the training data the model was trained on? Also how it translates to other specious? moreover, did you train one model per dataset, or is there a way to train all together?

---

> ### Author Rebuttal · Authors · 2024-08-04
>
> **Weakness: Formulation Clarify:**
>
> Thank you for your valuable feedback. Regarding the introduction of the R term in Eq. 7, we state the following:
>
> **Introduction of R:** R represents the posterior probability in the second part of the variational network. Its inclusion aims to enhance model expressiveness by capturing the underlying structure and parameter dependencies more effectively.
>
> **Expression of $q(\tau(z))$:** Detailed derivations in **Appendix, Equations 19-31** clarify that $q(\tau(z))$ is indeed Q(z), indicating that our model includes the KL divergence between R and Q, which is a crucial component. We will improve the presentation and explanation of these concepts in the revised manuscript to make them clearer.
>
> We will also enhance the overall presentation of the paper to reduce any confusion.
>
> **Question：**
>
> **Q1: Overlooked Sequence Features**
>
> Thank you for your question. The key sequence features we refer to involve the semantic aspects of genetic information, which are often overlooked by traditional distance-based methods. These methods typically rely on simplified one-hot encoding, which fails to capture the complex interactions and biological functions within sequences. In contrast, our approach utilizes pre-trained language models to extract richer representations from the sequence context, effectively capturing complex relationships and semantic information. This enhances the accuracy and flexibility of phylogenetic inference.
>
> **Q2: Differentiability of NJ Algorithm**
>
> Thank you for your question. You are correct that the Neighbor-Joining (NJ) algorithm is not differentiable. However, in our approach, NJ is only used to generate the initial tree topology and does not require differentiation. The overall model is trained end-to-end, with differentiation occurring in other parts of the model, not within the NJ algorithm.
>
> **Q3: Clarification on Section C.1:**
>
> Thank you for pointing out this issue. The notation $p(\tau(z))$ in the main text is incorrect and should be corrected to match the accurate representation in the **appendix**, specifically on **line 540**. Additionally, $q(\tau(z), B_\tau)$ corresponds to the distribution detailed in Eq. 19. We will ensure these notations are consistent and clear in the revised manuscript.
>
> **Q4: ELBO Metric:**
>
> Thank you for your insightful comment. You are correct that ELBO alone may not fully capture the model’s inference ability, which is why we introduced the Scoring function. Additionally, it is important to note that our baselines and recent methods also use ELBO and MLL, so we must use these metrics for fair comparison.
>
> **Q5: LLM Embeddings and Tree Construction:**
>
> Thank you for your question. The embeddings generated by the LLM are primarily for feature extraction, not for direct tree construction. The initial topology and the subsequent adjustments made by the first module are essential to ensure the embeddings are constrained to a Gaussian distribution via the encoder, as assumed in our model. Without this step, the embeddings might not converge properly, and the phylogenetic tree construction could be less accurate.
>
> **Q6: Scoring Function:**
>
> Thank you for your question. The scoring function maps each leaf node in the latent space to a score, representing their contribution to the model's overall performance. The function is jointly optimized with the ELBO objective to ensure the directionality of parameter updates during training. We minimize the scoring function to enforce low-rank representations in the matrix, which helps construct a well-structured initial tree, leading to faster convergence and improved accuracy.
>
> **Limitations:**
>
> **L1: Computational Resources**
>
> **1.1 Monte Carlo:** PhyloGen does not use traditional MCMC to sample from posterior distributions of tree topologies and branch lengths. Instead, we employ a Variational Bayesian approach [1], which significantly reduces computational demands by optimizing the variational lower bound to approximate the posterior, avoiding the inefficiencies and high sample requirements of MCMC in high-dimensional tree spaces.
>
> **1.2 NJ Algorithm:** NJ is used to construct the initial tree structure and requires some computational resources. It is executed during each iteration's initialization phase. While it adds to the computational load, this step ensures accuracy and stability, providing benefits that outweigh the computational costs and are not achieved by other methods.
>
> **1.3 DNABERT2:** This step is highly efficient. For instance, processing 100 sequences takes only 1.76s, and our benchmark datasets are much smaller, ensuring that embedding computation isn’t a bottleneck.
>
> [1] Zhang C, Matsen IV F A. Variational Bayesian Phylogenetic Inference[C]//ICLR (Poster). 2019.
>
> **L2: Training and Inference Performance:**
>
> You are correct in noting that we trained one model per dataset, similar to recent methods in the field. During inference, the model's performance is limited to the current dataset on which it was trained. We are exploring fully end-to-end approaches to address these limitations and improve generalization across different species and varying numbers of species.

---

> > ### Author Response · Authors · 2024-08-10
> >
> > Dear Reviewer,
> >
> > We sincerely appreciate your efforts and valuable feedback. If you are satisfied with our responses and our improvements, please consider updating your score.
> >
> > If you need further clarification, please don't hesitate to contact us. We are grateful for your time and look forward to your response!

---

> > ### Comment · Reviewer_4WGb · 2024-08-14
> > **Reviewers comment**
> >
> > Thank you for authors answer to clarify my questions. After reading the answer, I decide to keep my score, I think the idea is interesting but the paper presentations needs to be improved and parts of the methodology as well as the evaluation metrics needs to be clarified.

---

> > > ### Author Response · Authors · 2024-08-14
> > >
> > > Thank you for your thoughtful feedback and for considering our responses.
> > >
> > > We’re pleased you find the idea interesting and want to assure you that we’re committed to improving the paper’s presentation, methodology, and evaluation metrics, as you suggested.
> > >
> > >
> > > Your **positive opinion is crucial**, and we hope you **might reconsider your score** given our commitment to these improvements.
> > >
> > >
> > > Thank you again for your valuable insights.

---

### Official Review · Reviewer_XKPL · 2024-07-27

**Soundness:** 3
**Presentation:** 2
**Contribution:** 3
**Rating:** 6
**Confidence:** 2

**Summary:**

The authors propose a new method, PhyloGEN, for phylogenetic inference. The method is able to perform end-to-end variational inference in order to jointly optimize the tree topology and the branch lengths. To achieve this, the authors propose using a pre-trained genomic language model to extract genome embeddings, and the embeddings are then used to form an initial topology which is refined iteratively using the proposed topology and branch length learning modules.

**Strengths:**

- The paper makes a novel contribution to an important research topic. Unlike previous variational inference methods, the method does not require pre-generated topologies.

- The incorporation of a pre-trained genomic language model to the method is an interesting contribution and is something that will likely be increasingly explored in the future.

- The method is compared against a number of recent phylogenetic inference methods, and it achieves superior performance in terms of MLL and ELBO on standard benchmark datasets compared to the previous methods. In addition, experiments are performed to investigate the robustness of the proposed method and the diversity of the generated topologies, indicating that the method is consistent with the "gold standard" MrBayes approach. The authors also provide an ablation study to understand the impact of the various design choices in their method.

**Weaknesses:**

- The methods section (Section 3) is difficult to read in that variables are introduced without leading sentences, some notation is undefined, and design choices are introduced without clear motivation.
  - What is the motivation for $f_i$ and how is it defined in (2) if there are multiple children $f_j$?
  - The function $h$ is used before its definition.
  - Which function $h$ is the latter $h$ in the definition $h(x_i, x_j) = h(x_i, x_i - x_j, d_{ij}^2)$?
  - Why is max aggregation a good choice?
  - The scoring function $S$ is not described in enough detail. What does it output and how does it provide additional gradient information?

- Most experiments, in particular RQ2, RQ4, RQ5, RQ6, and RQ7 are performed on only one data set (DS1), making it difficult to draw meaningful conclusions. I would also consider the runtime an important metric, but it is measured only in conjunction with the robustness assessment on only one data set.

- The authors do not provide code for their method or experiments (although the code is promised to be released later). Without the code, considering the lack of detail in the methods section, it would be difficult to reproduce the proposed method.

- Minor:
  - The red and yellow colors are opposite to what they should be in the caption of Table 1.
  - There is a subsection numbered 3.1 in Section 4.
  - RQ7 is only referred to in the Appendix, and not in the main text.
  - Typos: "TreeEnocoder" (Figure 2 caption), "Metrtic" (Figure 3 title)

**Questions:**

- Could the authors explain that if MrBayes is considered the gold standard, and the proposed method is validated by comparing it against MrBayes in RQ2 and RQ3, why is the proposed method able to achieve such a big increase in MLL and ELBO while the other methods yield relatively similar results? Relatedly, do the resulting topologies look significantly different in comparison to the other methods?

- The proposed method has low standard deviations in Table 2 for multiple datasets. Is this due to the optimization being robust with respect to the random seed or could it not indicate that $Q(z)$ does not support multiple topologies? Do the MrBayes posteriors support multiple topologies?

- How does the Bipartition Frequency Distribution look for other methods like GeoPhy and ARTree?

- How does the runtime of MrBayes compare to the runtime of the proposed method?

- Can the authors hypothesize why a simple fully connected $S$ would work so well (Figure 3)?

- In the construction of the distance matrix, could the authors explain why using XOR makes sense since aren't the $z_i^*$ continuous? Moreover, would it make sense to try out other similar algorithms to neighbor-joining to initialize the topology?

- I would suggest that the authors move section 3.3 (with the derived gradients) to the Appendix, leaving space for a more detailed description of the various design choices.

**Limitations:**

The authors do not explicitly discuss the limitations of their work. As I am not an expert in phylogenetic inference, it is difficult to judge the assumptions made in the paper. Are there conditions that are required to be met for using a pre-trained language model for extracting embeddings, and is the assumption of the tree topology and branch lengths being conditionally independent reasonable? It would be helpful for the authors to elaborate on what they think the possible limitations of their own work are.

---

> ### Author Rebuttal · Authors · 2024-08-04
>
> **Weaknesses：**
>
> **W1: Clarity in Methods:**
>
> We acknowledge the reviewers' concerns about the presentation of the methods section. In response, we have made a thorough revision:
>
> 1. **$f_i$** enriches the node feature $i$ by integrating contributions from its children and its inherent traits. In Eq. 2, $f_j$ represents the features of child nodes, and for multiple children, we aggregate their contributions directly in Eq. 1.
> 2. And 3. **h Clarification:** We acknowledge the confusion caused by using the same symbol h for two conceptually different functions in the manuscript. The first $h(x_i, x_j)$ is an abstract representation used to describe a distance measure between two nodes. The second $h(x_i, x_i - x_j, d_{ij}^2)$ is a specific implementation of the first, calculating differences in the latent space between nodes x_i and x_j, evaluated using the formula $\sum (\| x_i - x_j \|^2 - d_{uv})^2$.
> 3. **Max Aggregation** selects the most significant features from parent and child nodes, ensuring the most prominent features are retained during propagation.
> 4. **Scoring Function S** maps each leaf node in the latent space to a score reflecting its contribution to the model's overall performance. Joint optimization with the ELBO objective ensures that parameter updates during training align with the intended direction.
>
> **W2: Ablation:**
>
> We chose DS1 because it is representative and commonly used in the field, ensuring comparability with other recent studies. Due to time constraints, we have included ablation results for some datasets in the attached file. We can add more if you feel it’s necessary. For a complete comparison of runtime, please refer to Tab. R2.
>
> **W3: Code:**
>
> Please see the General Response.
>
> **M1, M2, M3, M4:**
>
> Thank you for your correction. We have corrected the caption and subsection numbers accordingly. We will also explicitly reference RQ7 in the experimental section to better connect it with the appendix details.
>
> **Questions:**
>
> **Q1: Superior Performance Over MrBayes:**
>
> We appreciate your insightful inquiry. PhyloGen outperforms MrBayes and other methods on MLL and ELBO due to enhanced genetic sequence representations via a pre-trained genomic language model. This approach allows for a deeper utilization of genetic information, improving the accuracy of phylogenetic tree construction. For topology comparisons, we use metrics such as Simpson's Diversity Index and Frequency of the Most Frequent Topology. Results across multiple datasets, detailed in Tab. R1 shows PhyloGen's ability to generate diverse and balanced topologies, revealing subtle evolutionary relationships that traditional methods may miss while maintaining biological accuracy.
>
> **Q2: Low Stds:**
>
> Tab. 2's low stds demonstrate PhyloGen's robustness against random seed variations, maintaining low variance despite supporting diverse topologies. Our variance levels are comparable compared to recent methods like ARTree and PhyloGFN. In contrast, MrBayes, which uses a sampling-based approach, exhibits higher variance across varied topologies but does not match PhyloGen's performance even when considering maximum variance. Furthermore, MrBayes employs MCMC algorithms for estimating posterior distributions of phylogenetic trees, necessitating lengthy MCMC chains to achieve convergence. While parallel computing and evolutionary model optimization can mitigate some computational demands, they do not overcome the inherent efficiency and variance control limitations.
>
> **Q3: Bipartition Frequency Distribution:**
>
> Thank you for your inquiry. We have included a comparison of bipartition frequency distributions for GeoPhy, as detailed in Fig. R2. However, direct comparisons with ARTree are not feasible since its autoregressive generation method does not produce Newick files.
>
> **Q4: Runtime Comparison:**
>
> Thank you for your suggestion. We detailed this in Tab. R2.
>
> **Q5: Efficacy of a Simple FC Layer:**
>
> The simple FC layer's effectiveness can be attributed to the high-quality features provided by DNABERT2, which allow the basic architectures to handle complex tasks efficiently. Additionally, minimal parameters and appropriate regularization ensure model stability and convergence.
>
> **Q6: XOR in Distance Matrix:**
>
> Using XOR to construct the distance matrix effectively captures discrete nucleotide mismatches without requiring sequence alignment, directly measuring genetic variations. This approach aligns with recent methods like GeoPhy and VaiPhy. Additionally, we explored the UPGMA algorithm on DS1, and the results are summarized in Fig. R1 and the following table:
>
> |  | ELBO |
> | --- | --- |
> | NJ | -7005.98 |
> | UPGMA | -7115.95 |
>
> UPGMA assumes uniform evolution rates and is straightforward but limited to consistent-rate scenarios. In contrast, NJ doesn’t assume constant rates, offering more accurate reflections of evolutionary branching in uneven-rate organisms. Overall, NJ proves more effective in handling complex and uneven evolutionary data.
>
> **Q7: Suggestion:**
>
> Thank you for your suggestions; we will change them in the revised version.
>
> **Limitations:**
>
> Thank you for your inquiry. Our first use of pre-trained language models to extract embeddings in this domain is validated by our experimental results and not contested in the literature, and we expect that this could spark further discussion in the field. Regarding the dependence on tree topology and branch lengths, our assumptions align with recent methods such as ARTree, GeoPhy, and PhyloGFN, which are widely accepted. A detailed discussion of limitations can be found in the General Response.

---

> ### Comment · Reviewer_XKPL · 2024-08-09
>
> Thank you for answering my questions in detail and for providing a discussion on the limitations of your method. In light of this, I am slightly raising my score.
>
> I cannot comment on the standard practice of using only DS1 for the experiments, but given the various somewhat ad-hoc design choices in your method,  I would expect to see at least the experiments of Section 4.5 performed using more data sets in a final revision. I also hope the authors pay careful attention to the multiple presentation issues raised by me and the other reviewers.

---

> > ### Author Response · Authors · 2024-08-09
> >
> > Thank you for your valuable feedback and for raising your score. We are conducting the additional Ablation Study experiments and will share the results with you as soon as they are ready.
> >
> > We are also refining the manuscript to address the presentation issues raised by you and the other reviewers, aiming to enhance its clarity.

---

> > ### Author Response · Authors · 2024-08-10
> >
> > **Additional Experiments on Ablation Study:**
> >
> > Thank you for your feedback and for raising your score. I understand your concerns regarding the experiments in Sec. 4.5 and have conducted additional ablation studies on multiple datasets, summarized below:
> >
> > The rows represent different configurations: **Ours** (full model); **Ours w/o LN** (without layer normalization); **Ours-Hid=64** (reduced hidden dimensions); **Ours-Euclidean** and **Ours-Cosine** (replacing the custom distance matrix with Euclidean and cosine metrics); and **Ours-One-Hot** (using one-hot encoding for leaf nodes). The numbers in parentheses following each dataset (e.g., DS1 (27)) represent the number of species sequences.
> >
> >
> > **DS1 (27):**
> >
> > | Methods | ELBO (↑) | MLL (↑) |
> > | --- | --- | --- |
> > | Ours | -7005.98 | -6910.02 |
> > | Ours w/o LN | -7111.31 | -7187.40 |
> > | Ours-Hid=64 | -8081.97 | -7728.90 |
> > | Ours-Euclidean | -9284.57 | -9091.84 |
> > | Ours-Cosine | -8943.73 | -8757.81 |
> > | Ours-One-Hot | -10433.23 | -9282.87 |
> >
> > **DS2 (29):**
> >
> > | Methods | ELBO (↑) | MLL (↑) |
> > | --- | --- | --- |
> > | Ours | -26362.75 | -26257.09 |
> > | Ours w/o LN | -26443.78 | -26301.85 |
> > | Ours-Hid=64 | -27511.50 | -27373.58 |
> > | Ours-Euclidean | -29355.51 | -29240.52 |
> > | Ours-Cosine | -28265.33 | -28147.96 |
> > | Ours-One-Hot | -30954.67 | -30848.88 |
> >
> > **DS3 (36):**
> >
> > | Methods | ELBO (↑) | MLL (↑) |
> > | --- | --- | --- |
> > | Ours | -33430.94 | -33481.57 |
> > | Ours w/o LN | -33737.22 | -33481.57 |
> > | Ours-Hid=64 | -34400.24 | -34209.57 |
> > | Ours-Euclidean | -38067.04 | -37925.02 |
> > | Ours-Cosine | -35976.66 | -35823.20 |
> > | Ours-One-Hot | -39120.53 | -38989.87 |
> >
> > **DS4 (41):**
> >
> > | Methods | ELBO (↑) | MLL (↑) |
> > | --- | --- | --- |
> > | Ours | -13113.03 | -13063.15 |
> > | Ours w/o LN | -13395.25 | -13111.01 |
> > | Ours-Hid=64 | -13410.90 | -13115.28 |
> > | Ours-Euclidean | -15438.36 | -15270.67 |
> > | Ours-Cosine | -13417.38 | -13101.95 |
> > | Ours-One-Hot | -16760.54 | -16606.26 |
> >
> > **Analysis:**
> >
> > Using Euclidean or cosine distance metrics significantly lowers MLL, indicating their limitations in capturing complex evolutionary patterns. The one-hot encoding yields the lowest MLL, highlighting the superiority of our feature extraction module and pre-trained language model. Additionally, reducing hidden dimensions shows a clear performance drop, underscoring the importance of model capacity.
> >
> > We are conducting further experiments, and I will upload the full results as soon as they are available.

---

> > > ### Author Response · Authors · 2024-08-12
> > >
> > > **Additional Experiments on Ablation Study (Part 2):**
> > >
> > > Following our recent update, we have also completed additional ablation studies for datasets DS5 through DS8 to demonstrate our model's effectiveness further. The results support the conclusions drawn from the initial datasets. These results will be made available in the final manuscript for a comprehensive review.
> > >
> > >
> > >
> > > **DS5 (50):**
> > >
> > > | Methods | ELBO (↑) | MLL (↑) |
> > > | --- | --- | --- |
> > > | Ours | -8053.23 | -7928.4 |
> > > | Ours w/o LN | -8133.16 | -7947.05 |
> > > | Ours-Hid=64 | -8232.91 | -7953.48 |
> > > | Ours-Euclidean | -8884.67 | -8674.23 |
> > > | Ours-Cosine | -8550.97 | -8308.62 |
> > > | Ours-One-Hot | -9915.63 | -9728.71 |
> > >
> > >
> > > **DS6 (50):**
> > >
> > > | Methods | ELBO (↑) | MLL (↑) |
> > > | --- | --- | --- |
> > > | Ours | -6324.90 | -6330.21 |
> > > | Ours w/o LN | -6748.26 | -6341.30 |
> > > | Ours-Hid=64 | -7630.79 | -7337.23 |
> > > | Ours-Euclidean | -9039.58 | -8811.08 |
> > > | Ours-Cosine | -7822.54 | -7545.93 |
> > > | Ours-One-Hot | -9656.64 | -9437.25 |
> > >
> > >
> > > **DS7 (59):**
> > >
> > > | Methods | ELBO (↑) | MLL (↑) |
> > > | --- | --- | --- |
> > > | Ours | -36838.42 | -36838.42 |
> > > | Ours w/o LN | -37579.46 | -37085.76 |
> > > | Ours-Hid=64 | -37997.11 | -37571.84 |
> > > | Ours-Euclidean | -43182.38 | -42889.76 |
> > > | Ours-Cosine | -41446.14 | -41165.21 |
> > > | Ours-One-Hot | -45627.14 | -45358.70 |
> > >
> > >
> > > **DS8 (64):**
> > > | Methods | ELBO (↑) | MLL (↑) |
> > > | --- | --- | --- |
> > > | Ours | -8409.06 | -8171.04 |
> > > | Ours w/o LN | -8842.80 | -8358.32 |
> > > | Ours-Hid=64 | -9319.05 | -8911.28 |
> > > | Ours-Euclidean | -10128.19 | -9777.77 |
> > > | Ours-Cosine | -9812.72 | -9436.38 |
> > > | Ours-One-Hot | -13475.34 | -13187.75 |

---

> ### Author Response · Authors · 2024-08-12
> **Have our response addressed your concerns?**
>
> Dear reviewer XKPL,
>
>
> We have provided the complete results of the ablation study for the requested dataset (DS1-DS8) based on your suggestions.
>
>
> We sincerely appreciate all your valuable feedback to help us improve our work. **We believe that the issues raised by the reviewers have been addressed in the revision.** We hope that you will reconsider our submission of this response. If you are satisfied with our response and efforts, please consider updating your rating. If you need any clarification, please feel free to contact us. We are very pleased and look forward to hearing from you!
>
>
> Best regards,
>
> Authors

---

### Official Review · Reviewer_QAeZ · 2024-07-30

**Soundness:** 2
**Presentation:** 2
**Contribution:** 2
**Rating:** 5
**Confidence:** 2

**Summary:**

This paper presents PhyloGen, a novel approach for phylogenetic tree inference using pre-trained genomic language models and graph structure generation. PhyloGen aims to jointly optimize tree topology and branch lengths without relying on evolutionary models or equal-length sequence constraints. The method demonstrates superior performance and robustness across multiple real-world datasets compared to existing approaches.

**Strengths:**

Novel approach combining pre-trained genomic language models with graph structure generation for phylogenetic inference

Joint optimization of tree topology and branch lengths without typical constraints

Superior performance on benchmark datasets compared to existing methods

Robust to data changes and noise

Provides deeper insights into phylogenetic relationships

Computationally efficient compared to baselines

**Weaknesses:**

Limited discussion of potential limitations or failure cases

Lack of comparison to some recent methods in the field

Source code not provided

**Questions:**

How does PhyloGen perform on larger datasets with hundreds or thousands of species?

Have you explored using other types of pre-trained language models besides DNABERT2?

How sensitive is the method to hyperparameter choices?

What are the main computational bottlenecks of the approach?

Have you tested PhyloGen on simulated datasets where the true phylogeny is known?

**Limitations:**

While some robustness tests were performed (node additions and deletions), the paper doesn't explore all possible types of data noise or perturbations.

The paper doesn't deeply explore the interpretability of the model's decisions or the learned representations.

The paper doesn't extensively discuss how well the method generalizes to different types of genetic data or organisms not represented in the test datasets.

The method relies on pre-trained genomic language models (specifically DNABERT2), but doesn't explore how performance might vary with different pre-trained models.

---

> ### Author Rebuttal · Authors · 2024-08-04
>
> **Weakness:**
>
> **W1: Limitations:**
>
> Please see the General Response.
>
> **W2: Comparison with Recent Methods:**
>
> We appreciate the reviewer's concern regarding the comparison with recent methods. We believe that our manuscript already includes comparisons with several of the latest approaches, specifically GeoPhy (NIPS2023), PHyloGFN (ICLR 2024), ARTree (NIPS2023), and VBPI-GNN (ICLR2023), which have been influential in recent literature. However, we acknowledge the rapidly evolving nature of the field and invite the reviewer to suggest any additional methods that should be considered for comparison.
>
> **W3: Code:**
>
> Please see the General Response.
>
> **Questions:**
>
> **Q1: Performance on Larger Datasets:**
>
> Thank you for your question. We want to emphasize that PhyloGen has been tested on the same datasets as all recent methods, ensuring a fair evaluation. PhyloGen is designed to scale, yet there are currently no datasets available that contain hundreds to thousands of species. This is primarily because most phylogenetic analyses don’t require simultaneous processing of such extensive species counts; instead, research focuses on constructing subtrees for specific subsets. Phylogenetic methods fall into two categories: distance-based (UPGMA and NJ) and character-based(Maximum Parsimony (MP), Maximum Likelihood (ML), and Bayesian methods). The former is fast but limited, while the latter, although computationally intensive, offers greater precision. Review studies [1] suggest Bayesian is the most accurate, followed by ML and MP. Although PhyloGen offers significant speed improvements while maintaining accuracy, its performance on large datasets is still constrained by the inherent time complexity of these methods.
>
> [1] Hall, Barry G. "Comparison of the accuracies of several phylogenetic methods using protein and DNA sequences." *Molecular biology and evolution* 22.3 (2005): 792-802.
>
> **Q2 and L4: Pre-trained Language Models:**
>
> Thank you for your question. We selected the genome-specific foundation model DNABERT2 for our phylogenetic inference research. Although models like HyenaDNA and Nucleotide Transformer (NT) excel in long-sequence modeling, they are less apt for our specific needs. Upon your suggestion, we tested these models:
>
> | Method | MLL |
> | --- | --- |
> | DNABERT2 | -6910.02 |
> | HyenaDNA | -6918.63 |
> | NT | -6921.81 |
>
> DNABERT2 outperformed others, likely due to its specific optimization for genomic data. Our revised manuscript will detail these findings to justify our choice and showcase our method's flexibility.
>
> **Q3: Hyperparameter Sensitivity:**
>
> Thank you for your question. Our Experiment Analysis (see Section E.5 in Appendix A, Table 9) demonstrates PhyloGen's robustness across various hyperparameter settings, including Output Dimension, Hidden Dimension, and Layer Normalization. We also evaluated the impact of changing hidden dimensions with and without layer regularization. The results show that PhyloGen maintains robustness to hyperparameter choices.
>
> **Q4: Computational Bottlenecks:**
>
> We've detailed this in General Response's Limitations.
>
> **Q5: Simulated Datasets Tests:**
>
> Thank you for your insightful question. In phylogenetic analysis, "true phylogeny" often relies on theoretical assumptions derived from widely accepted biological software. As discussed in our introduction, these traditional methods require sequence alignment, which is time-consuming and computationally intensive. Additionally, our manuscript includes "PhyloTree Case Studies" (RQ6) and visualizations in Sec. E.6 demonstrates PhyloGen's ability to cluster biologically relevant species and highlight evolutionary relationships effectively. While defining true phylogenies in simulated datasets presents challenges, natural tree structures offer a more valid basis for inference. Furthermore, this setup is rare in standard references and baselines. We appreciate your interest, as it helps clarify our method's scope and potential enhancements. Future work will explore this area more thoroughly to validate PhyloGen's effectiveness.
>
> **Limitations:**
>
> **L1: Robustness:**
>
> Thank you for your comments. We've extended our tests to include scenarios such as genetic mutations. Here are the results from dataset DS1 under different settings:
>
> | Setting | Description | ELBO |
> | --- | --- | --- |
> | Setting 3 | Replaced sequences with alternatives from the same species | -7008.23 |
> | Setting 4 | Applied mutation rate of 5% | -7012.02 |
> | Setting 5 | Applied mutation rate of 10% | -7021.17 |
>
> The results show that the ELBO metric for Setting 3 is similar to the original result while Setting 4’s result is closer to the original than Setting 5. This indicates that our model maintains robustness under different types of data noise and perturbations, particularly in handling genetic mutations.
>
>
> **L2: Interpretability:**
>
>
> Thank you for your feedback. Our manuscript includes case studies in RQ6 and visualizations in Sec. E.6, demonstrating PhyloGen's application to real-world genetic data. These provide intuitive understanding and help enhance overall interpretability. Specifically, Fig. 7 illustrates the clustering and phylogenetic tree structure, effectively showcasing the representation results. Compared to similar studies, our approach provides a more extensive and comprehensive analysis, setting a high standard for interpretability within the field. We appreciate your feedback and will continue to refine this aspect.
>
> **L3: Data Generalizability:**
>
> Thank you for your question. Indeed, the benchmark datasets used by PhyloGen include a wide variety of organisms covering marine animals, plants, bacteria, fungi and eukaryotes. This ensures that our model has been tested in various biological environments, demonstrating its ability to generalize to different types of genetic data and organisms not specifically represented in the test dataset.
>
> **L4:Pre-trained Models:**
>
> Please see our response to Q2.

---

> > ### Author Response · Authors · 2024-08-10
> >
> > Dear Reviewer,
> >
> > We sincerely appreciate your efforts and valuable feedback. If you are satisfied with our responses and our improvements, please consider updating your score.
> >
> > If you need further clarification, please don't hesitate to contact us. We are grateful for your time and look forward to your response!

---

> ### Author Response · Authors · 2024-08-12
> **Summarized Response to the Unaddressed Concerns**
>
> Dear Reviewer QAeZ,
>
>
> We highly value your feedback and suggestions, which are crucial for us to review further and enhance our work. **We have revised our manuscript and addressed all your concerns.** We sincerely invite you to review our responses, hoping that it will help dispel any misconceptions about our work. Below is a summary of our response:
>
>
> - Performance on Larger Datasets: We clarified PhyloGen's scalability and ensured a fair comparison by testing it on the same datasets as other state-of-the-art methods.
> - Pre-trained Models and Hyperparameter Sensitivity: We justify the selection of DNABERT2 and demonstrate PhyloGen's robustness across different hyperparameter settings.
> - Interpretability and Generalizability: We highlight several case studies and visualizations that demonstrate PhyloGen's interpretability and generalizability in various biological contexts. We also expanded the robustness tests and addressed concerns regarding computational bottlenecks.
>
>
> In this revised version, we have sincerely and diligently responded to your valuable comments. **We trust that our responses have clearly addressed your questions and invite you to review them.**
>
>
> We are pleased to inform you that several reviewers have given **high scores**, with one even **raising their score** after reviewing our responses. Notably, the reviewer with **high confidence** recognized the strength and robustness of our approach. In light of this, we hope you reconsider your assessment and increase your score. If you have any further questions, we would be happy to address them.
>
>
> Best regards,
>
> Authors

---

### Author Rebuttal · Authors · 2024-08-05

**General Response:**

We are very grateful to the four reviewers for their insightful comments, which have significantly improved the quality and clarity of our manuscript.

**Summary:**

We are pleased that the reviewers recognized the highlights of our work, including the novel framework combining pre-trained genomic language models with graph structure generation for phylogenetic inference. This approach, which received praise from **all reviewers**, is expected to significantly impact the community and attract further exploration **(QAeZ, XKPL, 4WGb, MKto)**.

Our method achieves joint optimization of tree topology and branch lengths without typical constraints and does not require pre-generated topologies **(QAeZ, XKPL)**. It demonstrates state-of-the-art performance on real-world datasets **(QAeZ, XKPL, MKto)**. Extensive case studies and ablation experiments validate the robustness of our method and the diversity of the generated topologies **(QAeZ, XKPL)**. Additionally, our approach shows higher computational efficiency than baseline methods **(QAeZ)**. These results highlight our method's practicality and underscore its broad theoretical and applied potential **(QAeZ, XKPL)**.

We thank all reviewers for their insightful comments and for addressing some common concerns regarding the limitations and performance of our model:

**1. Limitations:**

While our model demonstrates outstanding performance on standard benchmarks, it may benefit from using more expressive Q(z) distributions or incorporating prior constraints to better capture complex dependencies and interactions in the latent space. Additionally, although the Neighbor-Joining (NJ) algorithm is effective for iterative tree construction, it is computationally intensive. We are exploring efficient data structures and parallel processing techniques to address this bottleneck. Furthermore, our model has primarily been applied to genomic data, and further research is needed to extend its applicability to diverse biological data, such as protein and single-cell data.

**2. Runtime Comparison:**

Despite these limitations, our model has performed exceptionally well in most evaluations and offers significant advantages in runtime and memory usage on the DS1 dataset:

|  | MrBayes | GeoPhy | PhyloGFN| ARTree | Ours |
| --- | --- | --- | --- | --- | --- |
| Runtime (H) | 22h46m | 8h10m | 20h40m | 62h21min | 6h53min |
| Memory (MB) | — | 1450.93 | 2341.80 | 2040.50 | 1051.11 |


**3. Source Code Availability:**

Due to time constraints, we have not yet fully organized all the code. The current version could be more organized, but we continue to refine it to meet the reviewers' needs. We have sent an anonymized link to the AC for reference.

**4. Thorough Ablation Study:**

We chose DS1 for our primary experiments because it is representative and commonly used in the field, ensuring comparability with other recent studies. Our original experiment design is both adequate and reasonable. In response to further curiosity and to provide additional details, we have included ablation results for additional datasets in the attached file. These are not to compensate for an insufficient design but to demonstrate the robustness of our method more comprehensively. If necessary, we can further supplement with more datasets.

Thank you again for your continued attention and valuable insights.

---

### Author Response · Authors · 2024-08-12
**General response to all the reviewers**

Dear Reviewers,


We greatly appreciate your effort and valuable feedback. In our response, we carefully illustrated and answered all your questions in detail. Additional experiments and analysis results are also provided as you had requested. In addition, we further clarified several unclear statements in the paper. We have incorporated all changes into the revised manuscript for your consideration. We hope your concerns have been addressed.


As you may know, unlike previous years, the discussion period this year can only last until August 13, and we are gradually approaching this deadline. We take it seriously and would like to discuss this with you during this time. We would be happy to provide more information based on your feedback or further questions.

If you are satisfied with our response, please consider updating your score. If you need any clarification, please feel free to contact us.

Best regards,
Authors

---

### Decision · Program_Chairs · 2024-09-25

**Decision:**

Accept (poster)

**Comment:**

The reviewers find the novelty of the method and its performance as notable strengths. (Minor) weaknesses include: the code being not yet mature enough to be released (it was only made available to the Area Chairs), missing comparisons to recent methods and otherwise limited empirical experiments, and some need for improvement in the presentation.

Overall, the recommendation is unambiguously in favor of accepting.

(As a final detailed comment: The terminology around "(pre-) aligned" vs "unaligned" sequences should be explicitly defined. To someone familiar with phylogenetic terminology, aligned sequences suggest multiple sequence alignment, which, correctly done, should in principle be more informative about phylogenetic relationships than unaligned sequences as one reviewer points out.)